# Traits of sub-kilometer F-region irregularities as seen with the Swarm satellites

Sharon Aol[1], Stephan Buchert[2], and Edward Jurua[1]

[1]Mbarara University of Science and Technology, Mbarara, Uganda
[2]Swedish Institute of Space Physics, Uppsala, Sweden

**Correspondence:** Sharon Aol (sharonaol@ymail.com)

**Abstract.** During the night, in the F-region, equatorial ionospheric irregularities manifest as plasma depletions observed by satellites and they may cause radio signals to fluctuate. In this study, the distribution characteristics of ionospheric F-region irregularities in the low latitudes were investigated using 16 Hz electron density observations made by a faceplate which is a component of the Electric Field Instrument (EFI) on board Swarm satellites of the European Space Agency (ESA). The study covers the period from October 2014 to October 2018 when the 16 Hz electron density data were available. For comparison, both the absolute ($\mathrm{std}(dN_e)$) and relative ($\mathrm{std}(dN_e)/N_e$) density perturbations were used to quantify the level of ionospheric irregularities. The two methods generally reproduced the local time, seasonal and longitudinal distribution of equatorial ionospheric irregularities as shown in earlier studies, demonstrating the ability of Swarm 16 Hz electron density data. A difference between the two methods was observed based on the latitudinal distribution of ionospheric irregularities where $\mathrm{std}(dN_e)$ showed a symmetrical distribution about the magnetic equator, while $\mathrm{std}(dN_e)/N_e$ showed a magnetic equator centered Gaussian distribution. High values of $\mathrm{std}(dN_e)$ and $\mathrm{std}(dN_e)/N_e$ were observed in spatial bins with steep gradients of electron density from a longitudinal and seasonal perspective. The response of ionospheric irregularities to geomagnetic and solar activities was also investigated using Kp index and solar radio flux index (F10.7), respectively. The reliance of $\mathrm{std}(dN_e)/N_e$ on solar and magnetic activity showed little distinction in the correlation between equatorial and off-equatorial latitudes, whereas $\mathrm{std}(dN_e)$ showed significant differences. With regard to seasonal and longitudinal distribution, high $\mathrm{std}(dN_e)$ and $\mathrm{std}(dN_e)/N_e$ values were often found during quiet magnetic periods compared to magnetically disturbed periods. The $\mathrm{std}(dN_e)$ increased approximately linearly from low to moderate solar activity. Using the high-resolution faceplate data, we were able to identify ionospheric irregularities of scales of only a few hundreds of meters.

**Keywords.** Low latitude ionosphere, Ionospheric irregularities, seasonal and longitudinal climatology

## 1 Introduction

Noticeable features in the low-latitude ionosphere are plasma density irregularities which occur after sunset in the F-region (Kil and Heelis, 1998). They may be identified as density decrease along satellite passes referred to as Equatorial Plasma Bubbles (EPBs) or range and frequency spread signatures on ionograms commonly called Equatorial Spread F (ESF) (Woodman and La Hoz, 1976; Burke et al., 2004). The generalized Rayleigh Taylor Instability (RTI) is the suggested mechanism which can

explain how ionospheric irregularities occur in the low latitudes (Woodman and La Hoz, 1976; Gentile et al., 2006; Portillo et al., 2008; Nishioka et al., 2008; Kelley, 2009; Schunk and Nagy, 2009). They may cause disruptions in trans-ionospheric radio signals of frequencies ranging from a few hundred kilohertz to several gigahertz, which in turn degrade the performance of communication and navigation systems (Kil and Heelis, 1998; Kintner et al., 2007).

There are many studies on the distribution characteristics of equatorial ionospheric irregularities (e.g., Kil and Heelis, 1998; Huang et al., 2002; Burke et al., 2004; Makela et al., 2004; Park et al., 2005; Su et al., 2006; Stolle et al., 2006; Kil et al., 2009; Dao et al., 2011; Xiong et al., 2012; Carter et al., 2013; Huang et al., 2014; Costa et al., 2014). Long-term observations of equatorial ionospheric irregularities have shown that their occurrence depends on various geophysical parameters including longitude, latitude, altitude, local time, season, solar activity and geomagnetic conditions (Kil et al., 2009). The dependence of
ionospheric irregularities on the geophysical parameters, however, remains a problem in modeling their variation for predictive purposes (Yizengaw and Groves, 2018). Therefore, further global-scale studies on the distribution characteristics of ionospheric irregularities and their dependence on various factors are still necessary. An interesting feature of these plasma irregularities is their scale sizes. They typically cover a variety of scale sizes, from a few centimeters to thousands of kilometers (Zargham and Seyler, 1989; Hysell and Seyler, 1998). The meter scales correspond to radio waves at HF and VHF frequencies where
irregularities are associated with Bragg scattering of radio waves and the spread F phenomenon (Woodman, 2009). The in situ measurements by satellites are normally suitable for analyzing the global aspects of the statistics of ionospheric irregularities. Depending on the sampling rate, spatial scales from about 100 m and larger can be seen. However, the orbital characteristics of satellite missions can limit the statistical coverage regarding altitude, latitude, local time, solar cycle phase, seasons, longitude, and aspect angle with respect to the geomagnetic field. Stolle et al. (2006) used magnetic observations made by the polar-
orbiting CHAllenging Minisatellite Payload (CHAMP) satellite for the years $2001-2004$ to study the irregularities. Multi-peak electron density fluctuations were not observed in the results presented by Stolle et al. (2006) due to the low sampling rate of the Planar Langmuir Probe (PLP) measurements. However, the CHAMP satellite's magnetic field data recorded at a frequency of 50 Hz showed irregularity structures as small as 50 m in scale size (Stolle et al., 2006). Multiple studies have also used high-resolution data sets when available to check the distribution characteristics of ionospheric irregularities and have been
able to resolve plasma density structures to smaller scales along satellite tracks (e.g, Lühr et al., 2014; Huang et al., 2014, etc).

The Launch of the first Earth observation constellation mission of the European Space Agency (ESA), i.e., Swarm, in November 22, 2013, generated new interests in the study of ionospheric irregularities. Each satellite is equipped with an Electric Field Instrument (EFI) in addition to other payloads. The EFI consists of LPs, and Thermal Ion Imagers (TII) (Knudsen et al., 2017). A number of studies have demonstrated the use of Swarm satellites for observations of ionospheric irregularities
(e.g, Buchert et al., 2015; Zakharenkova et al., 2016; Xiong et al., 2016; Xiong et al., 2016b; Wan et al., 2018; Yizengaw and Groves, 2018; Jin et al., 2019; Kil et al., 2019, etc). Most of these studies have used electron density measurements at a frequency of 2 Hz or 1 Hz made by the Langmuir Probes (LPs) on-board Swarm. Xiong et al. (2016) used Swarm 2-Hz electron density measurements to check the scale sizes of irregularity structures. They suggested that the structures have scale sizes in the zonal direction less than 44 km. The Swarm satellites have the capability of measuring electron density at an
even higher frequency of 16 Hz by determining the current through a faceplate. This plate is electrically isolated from the

satellite, negatively biased and located on the RAM side such that positive ions impact onto the relatively large surface of about $26 \times 26$ cm$^2$ with super-thermal velocity due to orbital motion (Buchert, 2016). As a result, the electron density can be readily estimated from the current at a relatively high rate of 16 Hz. The faceplate acts like a planar LP, however, without the possibility of sweeps and temperature measurements. Operation of the TII requires a bias value which turned out to be unsuitable for density estimation. Therefore, the 16 Hz density estimates are only provided when the TII is inactive (Buchert, 2016). Swarm can record ionospheric irregularities of scale lengths up to 500 m along their tracks using the 16 Hz electron density measurements. High-resolution data enables smaller scale structures to be identified in electron density (Nishioka et al., 2011). The 16 Hz electron density data from Swarm satellite has not yet been used to study traits of ionospheric irregularities. In the present study, we looked at the distribution characteristics of equatorial ionospheric irregularities using 16 Hz faceplate measurements of electron density. The study covers the period from October 2014 to October 2018 corresponding to the descending phase of solar cycle 24, when the 16 Hz electron density data was available. We show that the electron density measurements of Swarm faceplate can be applied to examine the characteristics of ionospheric equatorial irregularities at sub - kilometers scale lengths.

The rest of the paper is organized in the following order: The data and methods are presented in Sect. 2. The results are presented and discussed in Sect. 3. The summary and conclusions are presented in Sect. 4.

## 2 Data and Methods

The Swarm mission is made up of three same satellites (Swarm A, B, and C) with an orbital speed of around 7.5 km s$^{-1}$ in polar orbits. The Swarm satellites measure simultaneously the electron density ($N_e$), electron temperature ($T_e$), and spacecraft potential at a frequency of 2 Hz along the satellites' track with the LPs. The Swarm satellites also measure $N_e$ at a frequency of 16 Hz with a faceplate. The Plasma density is derived from the faceplate current assuming that it is carried by ions hitting the faceplate due to the orbital motion of the spacecraft (Buchert, 2016). Knudsen et al. (2017) provide more details on how electron density is derived from the LPs and TII. Using Swarm mission data collected from October 2014 to October 2018, orbit analysis was carried out to check on the status of the Swarm orbits. From the orbit analysis, by the end of October 2018, the longitudinal separation between Swarm A and C was about 1.4° corresponding to about 160 km distance at the equator, covering nearly the same local time sector with a time lag of about 1-10 seconds. The time lag between Swarm B and the lower pairs reached up to 8 hours. Swarm A and C were orbiting at an altitude of about 448 km (orbital inclination angle of 87.35°) above sea level over the low latitude region, and Swarm B was orbiting at an altitude of about 512 km (orbital inclination angle of 87.75°). In a day, the swarm satellites complete about 16 orbits with an average orbital period of about 91.5 min. Swarm satellites regress in longitude around 22.5° between orbital ascending nodes. Data sets measured by Swarm can be downloaded from `http://earth.esa.int/swarm`. The investigations done in this study are based on the 16 Hz $N_e$ faceplate data collected for the period of October 2014 to October 2018.

The identification criteria adopted for quantifying ionospheric irregularities have been a matter of concern. Some earlier studies (e.g, Kil and Heelis, 1998; McClure et al., 1998; Burke et al., 2003; Su et al., 2006; Kil et al., 2009; Dao et al., 2011,

etc) used relative plasma density disturbance to identify ionospheric irregularities while others (e.g, Lühr et al., 2014; Buchert et al., 2015; Xiong et al., 2016b, etc) took absolute density disturbance. However, Huang et al. (2014) used the 512 Hz Communication / Navigation Outage Forecasting System (C / NOFS) satellite's measurements of ion density and found that when the relative and absolute density disturbances are used independently, the likelihood of irregularities occurring and their variation with local time differ. The C / NOFS satellite was in a low tilt orbit, so the bubbles were sampled zonally. Important differences basing on latitudinal distribution of ionospheric irregularities using different criteria could not be addressed by Huang et al. (2014). The polar-orbiting Swarm satellites sample bubbles in a meridional direction and they give an opportunity to check the difference in the latitudinal distribution of irregularities using different identification criteria. The Swarm satellites cover mainly small aspect angles with respect to the magnetic field near the equator, while C/NOFS included mainly near field-perpendicular density variations. Chartier et al. (2018) used $N_e$ and Total Electron Content (TEC) measurements made by the LPs and GPS, respectively on board Swarm to test the relative and absolute perturbations in the detection of polar cap patches. In terms of seasonal distribution, they observed discrepancies between the two methods with relative disturbances showing more patches in winter than in summer. However, the study presented by Chartier et al. (2018) were limited to high latitudes.

For comparison purposes, two methods were also adopted for the polar-orbiting Swarm satellites to quantify the level of electron density irregularities in the low latitude region. In the first method, the 16 Hz $N_e$ measurements were passed through a 2-s (32 data points) running mean filter corresponding to a wavelength of about 15 km. From the original observations, the filtered data were subtracted to obtain the residual $dN_e = N_e - \overline{N}_e$; where $\overline{N}_e$ is the mean of $N_e$ at a 2-s interval. The standard deviation of the residuals which represents the density perturbation, std($dN_e$) was then calculated for every 32 data points. Basu et al. (1976) found that, on a global scale, absolute density perturbation equal to $1 \times 10^{-10}$ m$^{-3}$ represents the percentage occurrence of 140 MHz scintillations. Xiong et al. (2010) used absolute density disturbance thresholds of $5 \times 10^{10}$ m$^{-3}$ and $3 \times 10^{10}$ m$^{-3}$ respectively to identify density irregularity structures on CHAMP and GRACE observations. Wan et al. (2018) adopted absolute density perturbation $> 5 \times 10^{10}$ m$^{-3}$ to identify ionospheric irregularities from Swarm. Basing on the method used in the current study, only batches with std($dN_e$) greater than $0.25 \times 10^{10}$ m$^{-3}$ were considered to be significantly irregular and selected for extra processing and analysis. In the second method, std($dN_e$) was divided by $\overline{N}_e$ to obtain the relative perturbation, std($dN_e$)/$N_e$. There is no specific threshold definition to be used when std($dN_e$)/$N_e$ identifies irregularities (Huang et al., 2014; Wan et al., 2018). Kil and Heelis (1998) determined the likelihood of occurrence of relative disturbance $> 1\%(0.01)$ and $5\%(0.05)$ from Atmospheric Explorer-E (AE - E) satellite data. The AE - E satellite data was also used by McClure et al. (1998), but relative disturbances $> 0.5\%(0.005)$ were used to identify irregularities. To identify the occurrence of ROCSAT-1 irregularities, Su et al. (2006) and Kil et al. (2009) used a threshold of $0.3\%(0.003)$ for the relative disturbance. Huang et al. (2014) used high-resolution ion density measurements from the C / NOFS satellite and took relative perturbation $> 1\%(0.01)$. Wan et al. (2018) considered relative perturbation values larger than 20%. In the current study, only batches with std($dN_e$)/$N_e > 0.01$ were considered to be significantly irregular and used for further analysis basing on the methods adopted. Here, we mostly dealt with small scale ionospheric irregularities which are relevant for L-band scintillations. It is essential to note that these small-scale irregularities are not autonomous from those of large-scale ionospheric irregularities and they were

not differentiated. The only satellite sampling the bottom-side, at altitudes below 300 km, has so far been the Atmosphere Explorer-E (AE-E) (Kil and Heelis, 1998). With the AE-E satellite, irregularities with relatively small amplitudes were seen without clearly developed EPBs. At altitudes above 350 km addressed by nearly all other studies, smaller scale irregularities are usually embedded in EPBs. Automatic detection algorithms used in previous works do not discriminate between depletions (EPBs) and irregular multi-peak variations, with the exception of Wan et al. (2018), whose algorithm determined a depletion amplitude. The results are presented and discussed in the following section.

## 3 Results and Discussions

The high-resolution Swarm faceplate $N_e$ data were used to characterize ionospheric irregularities using procedures described in Sect. 2. Figure 1 shows examples of $N_e$ results for arbitrary passes of Swarm A and C on 2014-10-06 and 2015-07-03,

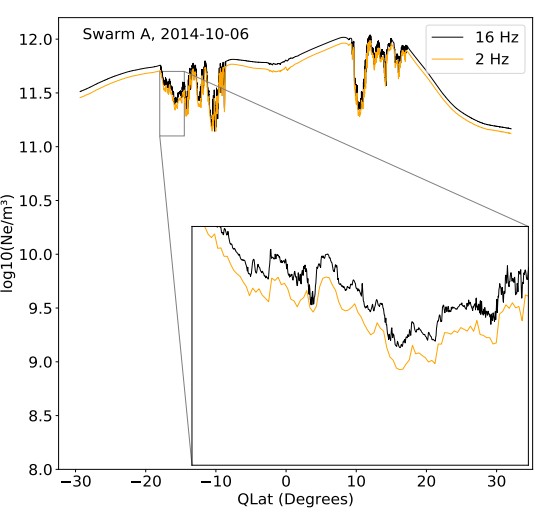
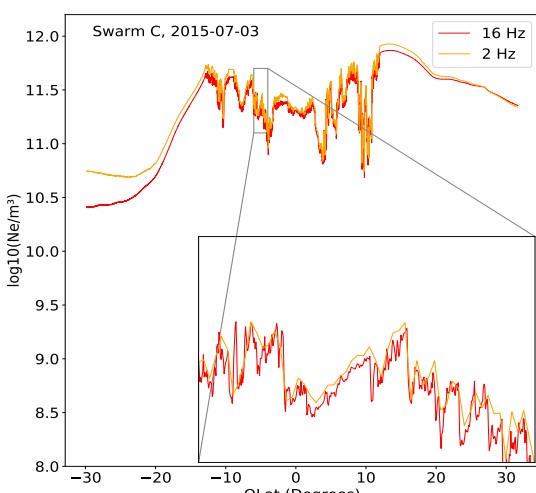

**Figure 1.** Comparison of 2 Hz LP data and 16 Hz faceplate data for Swarm A and C on 2014-10-06 and 2015-07-03, respectively.

respectively from the LP and faceplate to highlight the capability of the 16 Hz $N_e$ data for observations of irregularity density structures. In Fig. 1, the orange curve is the time series of the 2 Hz LP data, while the black and red curves are the time series of the 16 Hz faceplate data for Swarm A and C, respectively. Both the 16 Hz and 2 Hz $N_e$ data show large density depletion along the satellite tracks concentrated at about $\pm15°$ quasi-dipole latitude (QLat). The 16 Hz $N_e$ data were able to capture fluctuations in $N_e$ just as the 2 Hz data. However, smaller scale electron density depletions in $N_e$ cannot be verified with the

low resolution 2 Hz data as shown in the zoomed-in sections in Fig. 1. One of the drawbacks associated with the 16 Hz $N_e$ data, as mentioned earlier, is that it is only recorded when the TII is inactive. Therefore, to check data availability, Table 1 summarizes the number of satellite passes per year for which 16 Hz $N_e$ data were recorded. We used all the passes available

**Table 1.** Summary of yearly total Swarm satellite passes over the low latitude region for which 16 Hz data was recorded.

| Satellite | Total Swarm satellite passes per year | | | | | Total passes per satellite |
|---|---|---|---|---|---|---|
| Year | 2014 | 2015 | 2016 | 2017 | 2018 | |
| Swarm A | 711 | 7,158 | 6,670 | 6,520 | 4,242 | 25,301 |
| Swarm C | 891 | 1,390 | 8,208 | 7,101 | 5,748 | 23,068 |
| Swarm B | 1,127 | 7,836 | 7,522 | 7,017 | 2,895 | 26,397 |

as summarized in Table 1 and later realized that data accumulation for the period of study was enough for a climatology study.

Examples of Swarm's encounters with ionospheric irregularities are presented in Fig. 2. Figure 2 panel (a) shows multiple $N_e$ depletions occurring between about $\pm10° - \pm20°$ QLat. From Fig. 2 panels (b) and (c), large values of both $\mathrm{std}(dN_e)$ and $\mathrm{std}(dN_e)/N_e$ often occur in locations of large depletions in $N_e$ at the EIA crests close to the quasi-dipole equator, but also at the bottom of large scale bubbles. Based on the thresholds defined in Sect. 2 ($\mathrm{std}(dN_e) > 0.25 \times 10^{10}$ m$^{-3}$ and $\mathrm{std}(dN_e)/N_e > 0.01$) to identify plasma density structures, these depletions are ionospheric irregularities. The RTI is the most

known mechanism that causes irregularities in low latitudes (Kelley, 2009; Kintner et al., 2007). The lower ionospheric layer declines rapidly during the night compared to the top layer. This creates a sharp vertical gradient of plasma density directed upwards, contrary to the gravitational force's direction of action. For such unstable arrangement, irregularities in the F - region at the bottom intensify and drift up, creating more complex plasma structures that extend to higher altitudes along magnetic field lines (Woodman and La Hoz, 1976; Abdu, 2005; Kelley, 2009). In general, ionospheric irregularities are more intense at

the Equatorial Ionization Anomaly (EIA) belts ($\pm15°$ QLat) than at the geomagnetic equator as observed in Fig. 2. However, from Fig. 2, the event presented for Swarm A and C on 2015-03-07 shows high values of $\mathrm{std}(dN_e)/N_e$ even at the magnetic equator. Huang et al. (2014) also observed that the relative and absolute perturbations were both able to capture fluctuations in ion density measurements made along C / NOFS tracks during $2008-2012$ in the zonal direction. However, it was not possible to see a more detailed latitude distribution using C / NOFS satellite because it covered a small latitude range of about $\pm13°$

due to its low inclination angle of about $13°$. The local time distribution characteristics of ionospheric irregularities were also determined and the results are presented and discussed in the following subsection.

### 3.1   Local Time Distribution of Ionospheric Irregularities

It is known from many studies (e.g, Kil and Heelis, 1998; Burke et al., 2004; Su et al., 2006; Stolle et al., 2006; Dao et al., 2011; Huang et al., 2014; Xiong et al., 2016b; Wan et al., 2018, etc) that ionospheric irregularities in the low latitudes occur after

25 sunset. Here, we also check the local time dependence of the ionospheric irregularities identified on the $N_e$ data from the Swarm faceplate to compare with previous results. Figure 3 presents the percentage occurrence of equatorial ionospheric irregularities

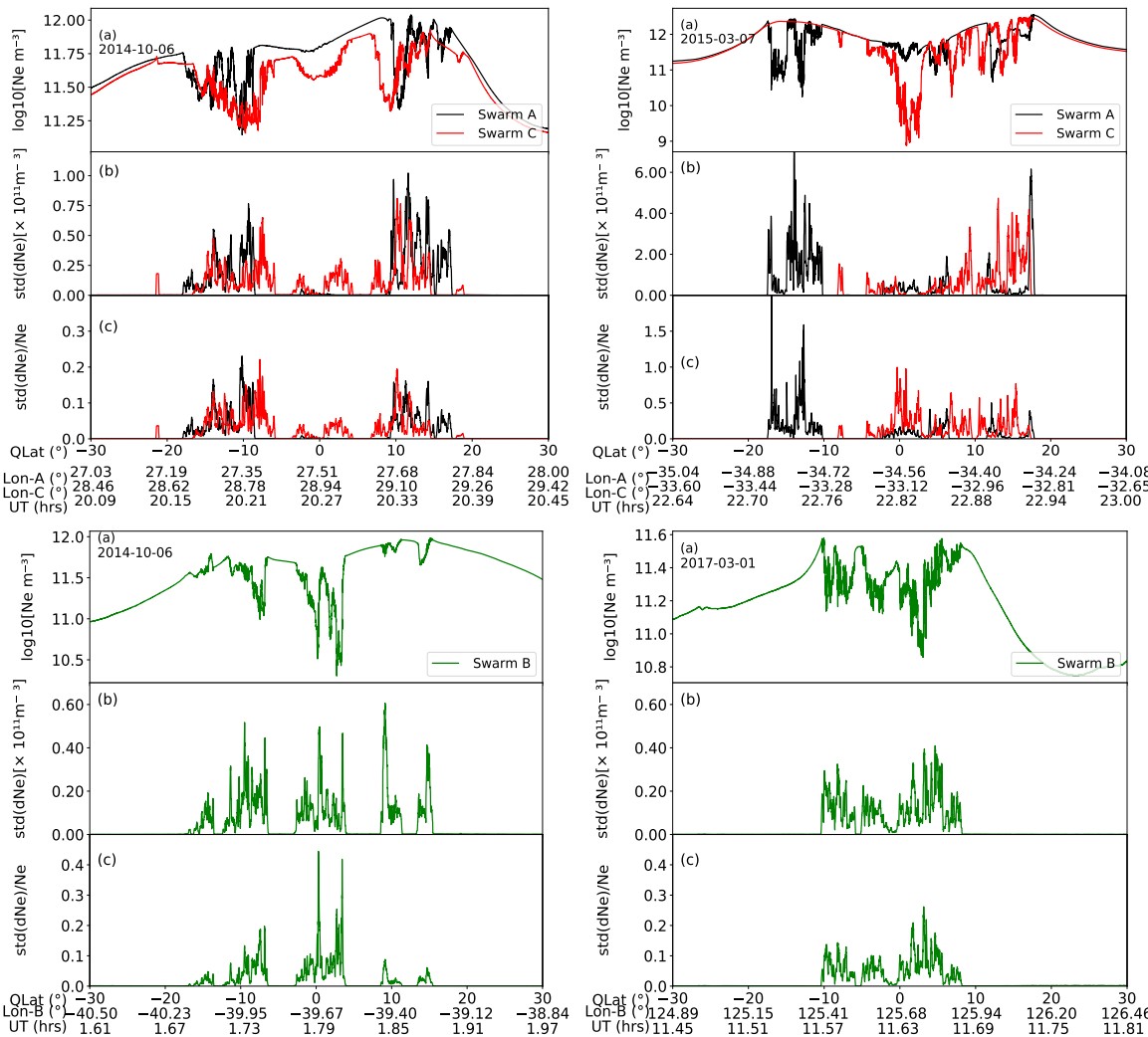

**Figure 2.** Irregularity structures observed by Swarm A, C, and B. Panels (a) to (c) represent electron density ($N_e$) variation at 16 Hz in logarithmic scale, absolute ($\mathrm{std}(dN_e)$) and relative ($\mathrm{std}(dN_e)/N_e$ ) electron density perturbations, respectively as functions of QLat, Geographic longitude (Lon), and Universal Time (UT).

as a function of local time based on (a) $\mathrm{std}(dN_e)$ and (b) $\mathrm{std}(dN_e)/N_e$. Using 16 Hz $N_e$ data accumulated during the period of study, the seasonal dependence of local time distribution of ionospheric irregularities was also examined by grouping all the data into different seasons corresponding to March Equinox (Feb-Mar-Apr), June Solstice (May-Jun-Jul), September Equinox (Aug-Sep-Oct), and December Solstice (Nov-Dec-Jan). The number of irregularity structures was determined per one hour

5  local time bin by counting the number of irregularity structures in a bin divided by the total number of observations.

**(a)** std($dN_e$)

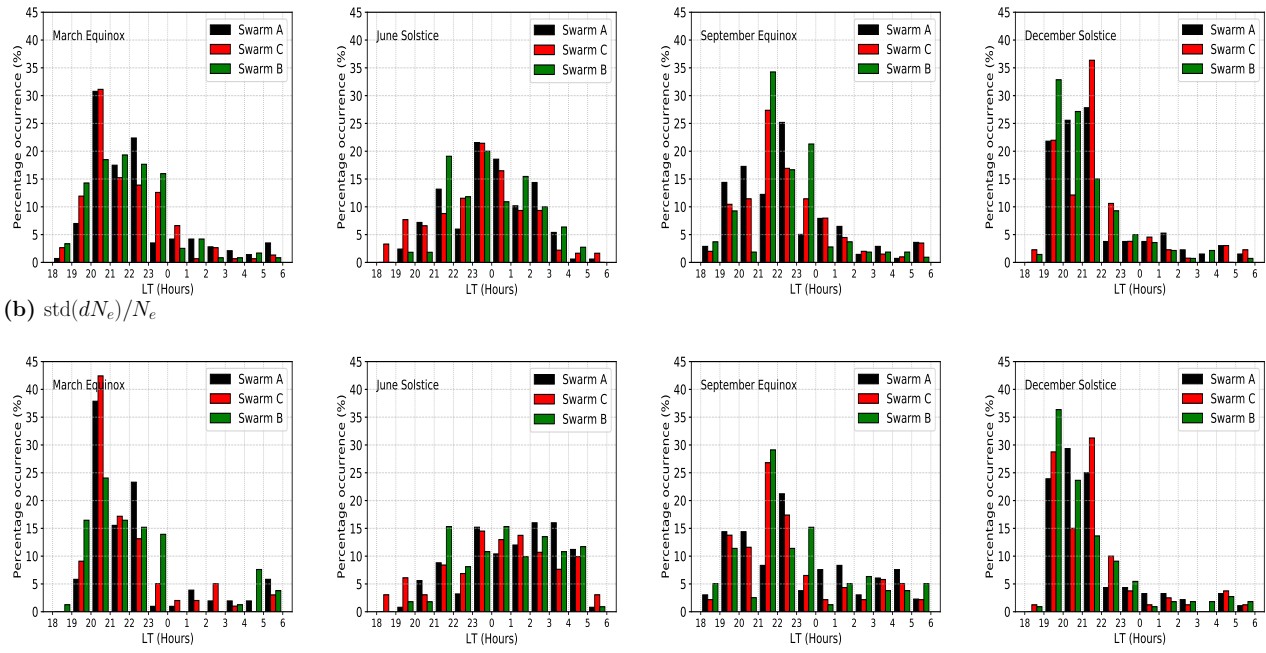

**Figure 3.** Histograms showing the percentage occurrence of (a) std($dN_e$)$> 0.25 \times 10^{10}$ m$^{-3}$ and (b) std($dN_e$)/$N_e > 0.01$ as a function of local time (LT) for the period from October 2014 - October 2018 for Swarm A, B, and C. Each panel presents a season.

As mentioned earlier, ionospheric irregularities in the low latitudes are nighttime phenomena and therefore, the analysis was restricted to the time period from 1800 LT-0600 LT. From Fig. 3, it is seen that irregularities occur from 1800 LT to 0600 LT as expected, irrespective of the method used. In Fig. 3, the highest percentage occurrence is observed in the equinoxes and December solstice, where the percentage increases faster between 1800 LT and 2000 LT, attaining a maximum at about 2100

LT and then decreases gradually till the morning hours. The increase in percentage occurrence from 1800 LT to 2100 LT can be attributed to increased eastward electric fields produced by the eastward thermospheric wind's electrodynamic interaction at the day-night terminator around the dip equator with the geomagnetic field (Rishbeth, 1971; Su et al., 2009). The increase in the electric field to the east causes the night-side ionosphere to rise to higher altitudes where RTI is favored and this increases the occurrence of ionospheric irregularities (Fejer et al., 1999; Abdu, 2005; Su et al., 2009). The percentage occurrence of

ionospheric irregularities is low in the June solstice and the percentage increase is slower with a wide plateau extending past midnight. According to Su et al. (2009), a late reversal of zonal drift associated with a small upward vertical post-sunset drift occurring at positive magnetic decline lengths in June solstice significantly inhibits irregularities. For the case of std($dN_e$), the percentage occurrence reduces towards morning hours, while std($dN_e$)/$N_e$ maintains a high percentage occurrence past midnight in June Solstice. The percentage occurrence trend of std($dN_e$)-based irregularities is like that of Kil and Heelis

(1998), Palmroth et al. (2000), Burke et al. (2004), Su et al. (2006), Stolle et al. (2006), Su et al. (2009), Xiong et al. (2016b),

Wan et al. (2018), etc. The $\text{std}(dN_e)/N_e$ shows a nearly similar trend in percentage occurrence as for $\text{std}(dN_e)$ but with high occurrence post-midnight in June solstice. Increase in post-midnight irregularities quantified by relative perturbations has also been observed by Huang et al. (2011), Huang et al. (2012), Huang et al. (2014) and Dao et al. (2011) who used ion density measurements made by C / NOFS. The mechanisms that generate these post-midnight irregularities are still unknown and widely debated. Two mechanisms to explain the occurrence of post-midnight irregularities have been suggested. One mechanism is the seeding of the RTI by atmospheric gravitational waves from below into the ionosphere, while the other mechanism is the elevation of the F-layer by the thermosphere's meridional neutral winds, which may be connected with the thermosphere's highest midnight temperature (Otsuka, 2018, and references therein).

The Global Positioning System - SCINtillation Network and Decision Aid (GPS - SCINDA) has often been used as one of the tools for measuring variations in radio signals' amplitude and phase. In the absence of GPS - SCINDA, many studies (e.g, Basu et al., 1999; Yang and Liu, 2015; Yizengaw and Groves, 2018) have shown that the rate of change of TEC index ($ROTI$) derived from Global Navigation Satellite System (GNSS) Total Electron Content (TEC) can be used as a proxy for quantifying scintillations. The $ROTI$ is defined as the standard deviation of Rate of Change of TEC ($ROT$) (Pi et al., 1997). Numerous studies have widely discussed these indices (e.g, Pi et al., 1997; Basu et al., 1999; Zou and Wang, 2009; Zakharenkova et al., 2016; Kumar, 2017; Yizengaw and Groves, 2018). We adopted $ROTI$ to compare the ground-based local time variations of irregularities/scintillations over different International GNSS Service (IGS) stations installed along the low latitude region with the variations presented in Fig. 3. Xiong et al. (2016a) and Wan et al. (2018) developed an algorithm to determine depletion amplitudes and they showed that large depletion amplitudes of EPBs are relevant for causing radio signal disruptions at L-band frequency. On the other hand, some previous studies have also stated the relevance of the small-scale $N_e$ structures for causing L-band scintillations (e.g, Spogli et al., 2016; Lühr et al., 2014; Bhattacharyya et al., 2003; Rao et al., 1997). Sharma et al. (2018) suggested that small-scale structures may be abundantly available within the largely depleted EPBs, becoming the cause of L-band scintillation. Therefore, the results presented by Xiong et al. (2016a) and Wan et al. (2018) could be explained because deep EPBs involve steep density gradients and large background density which would create the small-scale irregularities. From the method used in our study, we mainly focused on irregularities of wavelength of about 15 km along the satellite track and this is within the range of applicable Fresnel scales which is theoretically relevant as a cause of L-band scintillations. The IGS stations considered in this study are shown in Fig. 4 as red stars. The details of the IGS stations used are summarized in Table 2. To find $ROTI$, only signals from GPS satellites with elevation angle higher than $25°$ over each independent station were considered to reduce the multipath effects. The $ROTI$ values $> 0.5$ TECU/min (1 TECU $= 10^{16}$ el/m$^2$) were classified as irregularities/scintillations (Ma and Maruyama, 2006). Figure 5 presents the percentage occurrence of $ROTI > 0.5$ TECU/Min in 1-hour local time bins for the different IGS stations and seasons. It is important to note that RIOP did not have TEC data in June solstice and September equinox as seen in panels (b) and (c) of Fig. 5. In general, the trend followed by local time distribution of $ROTI$ seems to closely agree with that of $\text{std}(dN_e)$ and $\text{std}(dN_e)/N_e$ in the equinoxes and December Solstice. As expected the percentage occurrence of ionospheric irregularities is higher mainly for the IGS stations in the African longitude even in June Solstice (Yizengaw et al., 2014). The percentage occurrence in June Solstice is generally small, comparable to that observed in Fig. 3, with a broad

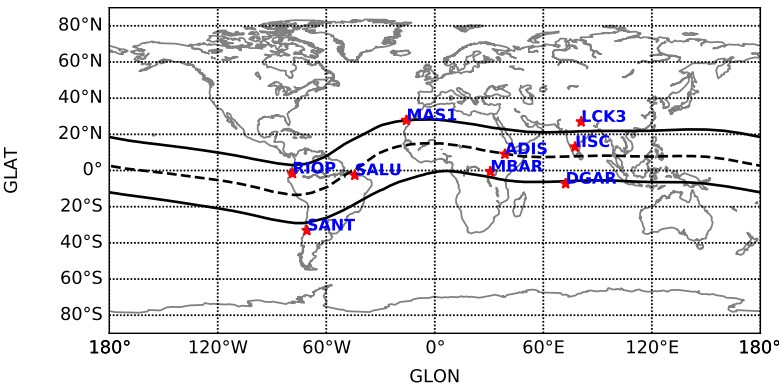

**Figure 4.** Map showing the location of IGS stations (red stars) considered in this study. The black dotted line represents the magnetic equator, while at about $\pm15°$ magnetic latitude the black solid lines represent the EIA belts.

**Table 2.** Coordinates of IGS stations used in this study.

| Station(Code) | Coordinates | | |
|---|---|---|---|
| | Geog. lat (°) | Geog. lon (°) | Mag. lat (°) |
| Mbarara (MBAR) | -0.60 | 30.74 | -10.2 |
| Addis Ababa (ADIS) | 9.04 | 38.77 | 0.18 |
| Maspalomas (MAS1) | 27.76 | -15.63 | 15.63 |
| Riobamba (RIOP) | -1.65 | -78.65 | 10.56 |
| Santiago (SANT) | -33.15 | -70.67 | -19.52 |
| Sâo Luis (SALU) | -2.59 | -44.21 | -0.25 |
| Diego Garcia Island (DGAR) | -7.27 | 72.37 | -16.89 |
| Bangalore (IISC) | 13.02 | 77.57 | 5.34 |
| Lucknow (LCK3) | 26.91 | 80.96 | 20.59 |

plateau extending post-midnight. However, the enhanced post-midnight irregularities seen in Fig. 3(b) for $\mathrm{std}(dN_e)/N_e$ for June Solstice are not observed in the LT trend of ROTI. Therefore, the LT dependence of percentage occurrence of ionospheric irregularities quantified using $\mathrm{std}(dNe)$ closely follows the same trend as that of ROTI for all seasons. The seasonal and longitudinal distribution of ionospheric irregularities is presented in the following subsection.

5    **3.2    Seasonal and Longitudinal Distribution of Ionospheric Irregularities**

The Swarm mission's 16 Hz $N_e$ data collected over the 5-year period $(2014-2018)$ has a credible global spatial and temporal coverage that is sufficiently good for examining the seasonal and longitudinal distribution of ionospheric irregularities in the low latitudes. To check the seasonal and longitudinal variation of ionospheric irregularities, we concentrate on satellite

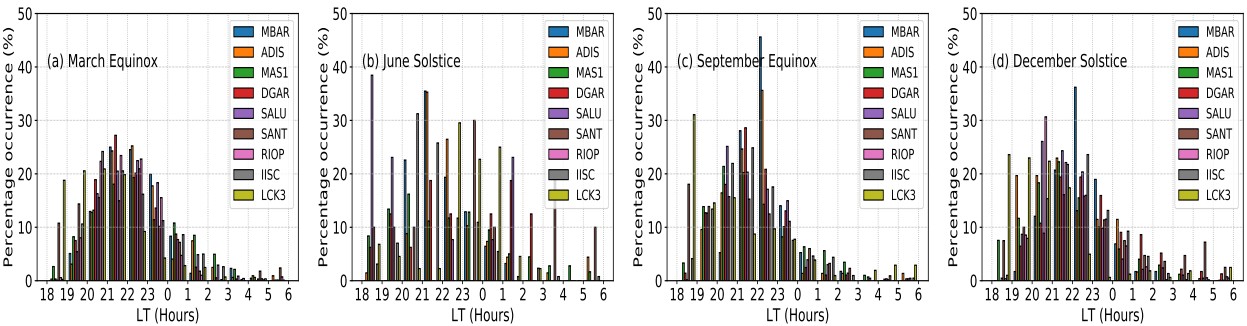

**Figure 5.** Percentage occurrence of $ROTI > 0.5$ TECU/Min in 1 hour LT bins for different stations (see legend).

passes within the local time range, $1800 - 0600$ LT. The $N_e$ data for the period of study was divided into four seasons as described in Sect. 3.1. Swarm equator crossings spanning the range of $-40° - +40°$ were considered since the study narrows down to the low latitude region. The std($dN_e$) and std($dN_e$)/$N_e$ were calculated in bins of $3° \times 4°$ resolution in geographic latitude and longitude. The occurrence rate of ionospheric irregularities does not always correspond to the highest amplitude of irregularity structures from the results presented by Wan et al. (2018). Therefore, here we concentrate on the magnitude of ionospheric irregularities other than the rate of occurrence. Zakharenkova et al. (2016) compared Swarm A and B 1-s $N_e$ data and revealed satellite-to-satellite differences related to altitude, longitude, and local time. Here, we also show the results for all the three satellites separately. Figure 6 shows the seasonal and longitudinal distribution of std($dN_e$) during the period of study in geographic coordinates, while Fig. 7 presents that of std($dN_e$)/$N_e$ for Swarm A, C, and B independently in the first, second, and third columns, respectively. The different seasons are shown in the four panels from top to bottom.

The first noticeable feature in Fig. 6 and Fig. 7 is that almost all the irregularities occur within the EIA belts between about $\pm15° - \pm20°$ magnetic latitudes. However, Fig. 6 shows that absolute variations of std($dN_e$) are observed with a gap of low values at the magnetic equator, while in Fig. 7 maximum values of std($dN_e$)/$N_e$ extend from the northern crest to the southern crest, including the magnetic equator. A clear picture of the density variations across the magnetic equator is seen in a scatter plot of the irregularities as a function of latitude as shown in Fig. 8. Some earlier studies (e.g, Burke et al., 2004; Su et al., 2006, etc) observed a normal-like distribution that peaks at the quasi - dipole equator and gradually decreases towards higher latitudes, reaching a minimum at around $\pm30°$ QLat, while others observed ionospheric irregularities concentrated around the northern and southern EIA belts (e.g, Liu et al., 2005; Stolle et al., 2006; Carter et al., 2013, etc). Only a few losses of the GPS tracks have been seen on the quasi-dipole equator (e.g, Buchert et al., 2015; Xiong et al., 2016a; Wan et al., 2018). Consequently, the variations in electron density at the quasi - dipole equator are relatively harmless to the GPS, the high-risk region being the high-density bands, north and south (Buchert et al., 2015).

Furthermore, there are differences between Swarm A / C and B in seasonal and longitudinal irregularity distribution. Swarm B shows the lowest values of both std($dN_e$) and std($dN_e$)/$N_e$ compared to A and C. A similar observation was made by

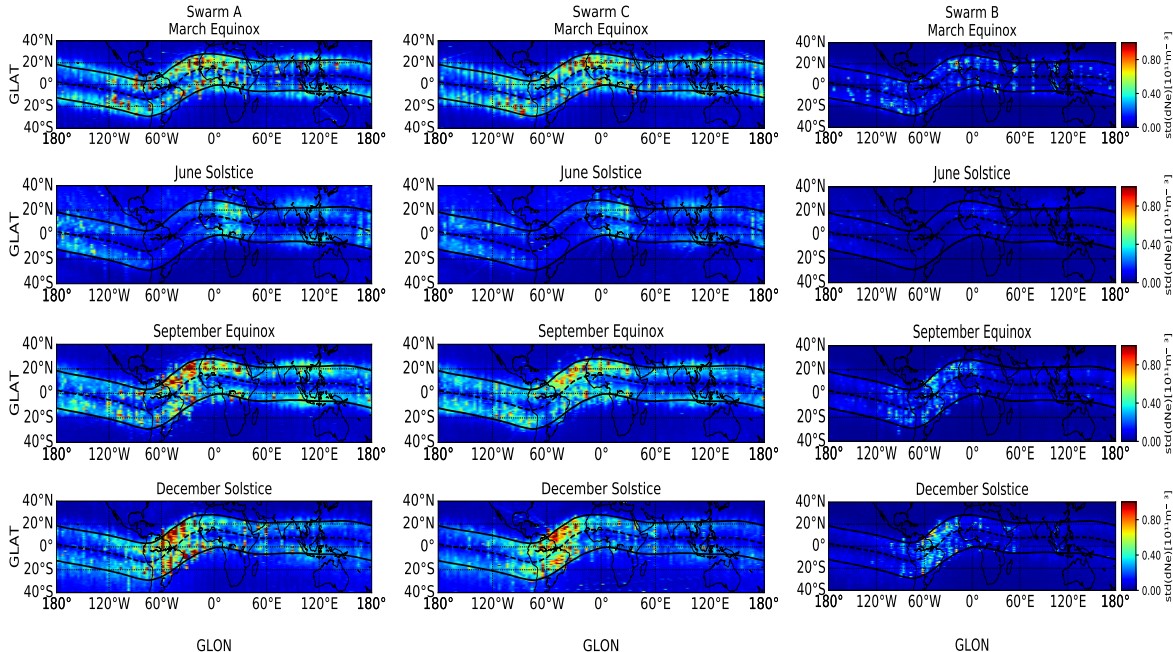

**Figure 6.** Absolute electron density perturbation ($\mathrm{std}(dN_e)$) separated into four seasons (March/September equinox and June/December solstice) from October 2014 to October 2018 for Swarm A, C, and B. The black dotted line represents the magnetic equator, while the black solid lines represent the EIA Belts at about $\pm 15°$ magnetic latitude. For each panel of a season, the color scales represent $\mathrm{std}(dN_e)$.

Zakharenkova et al. (2016) who compared the seasonal and longitudinal variation of ionospheric irregularities for only Swarm A and B during the years $2014-2015$ using the 1-s $N_e$ LP data. The differences observed between Swarm A/C and Swarm B can be explained by the altitude and local time separation between the satellites as Swarm B flies at a higher altitude and always crosses the post-sunset sector later than A and C.

In terms of seasons, high values of $\mathrm{std}(dN_e)$ and $\mathrm{std}(dN_e)/N_e$ are observed during the equinoxes at all longitudes especially in the African-Atlantic-South American regions. During June solstice, moderate values occur mostly in the African sector and the lowest values occur in the Atlantic-South American sector. During December solstice, high values are observed in the Atlantic-American sector. From the Indian Ocean to central Pacific sectors where the magnetic field declination is low, no satellite detected many intense ionospheric irregularities in solstice seasons and in September equinox. Overall, the seasonal

and longitudinal irregularity distribution shown in Fig. 6 and Fig. 7 is consistent with earlier studies irrespective of the criteria adopted (e.g., Su et al., 2006; Huang et al., 2001; Burke et al., 2004; Park et al., 2005; Huang et al., 2014; Zakharenkova et al., 2016; Wan et al., 2018). The RTI is known to intensify after sunset, causing severe irregularities when the day-night terminator is aligned with the plane of the magnetic field that occurs in the equinox (Tsunoda, 1985; Burke et al., 2004; Gentile et al., 2006; Yizengaw and Groves, 2018).

One of the challenges has been explaining the mechanism governing the longitudinal distribution of irregularities. Tsunoda

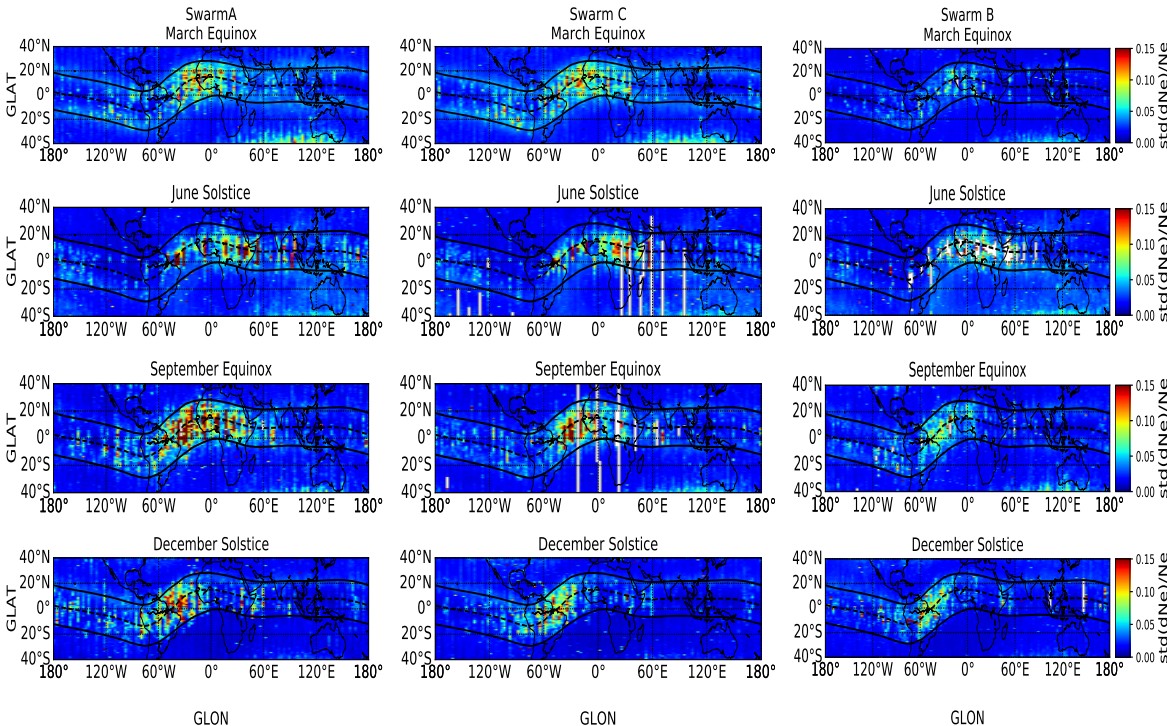

**Figure 7.** Relative electron density perturbation $(\mathrm{std}(dN_e)/N_e)$ separated into four seasons (March/September equinox and June/December solstice) from October 2014 to October 2018 for Swarm A, C, and B. The black dotted line represents the magnetic equator, while the black solid lines represent the EIA Belts at about $\pm 15°$ magnetic latitude. For each panel of a season, the color scales represent $\mathrm{std}(dN_e)/N_e$.

(1985) proposed a model based on the magnetic declination to explain the distribution of ionospheric irregularities. However, this model could not explain the high occurrence of irregularities in June solstice over the African longitude. The longitudinal distribution of irregularities has also been attributed to gravity waves originating from the thermosphere (Yizengaw and Groves, 2018, and references therein). Yizengaw and Groves (2018) also added that the intertropical convergence zone (ITCZ) position, which are sources of gravity waves, may explain the longitudinal irregularity dependence observed. Kil et al. (2004) suggested that the longitudinal distribution at EIA latitudes of absolute electron density affects the occurrence of irregularities. Using DMSP data, Huang et al. (2001), Huang et al. (2002), and Burke et al. (2004) showed that the pattern of precipitation of the inner radiation belt's energetic particles explains the pattern of irregularities.

Among other parameters, the growth rate of equatorial ionospheric irregularities is controlled by the electron density gradient. Ionospheric irregularities in the equatorial and low latitudes can cascade upwards and along the magnetic field lines to the EIA belts characterized by high background $N_e$ and steep gradients in density (Muella et al., 2010). From both local time and longitudinal perspectives, Wan et al. (2018) confirmed that the depletion amplitudes of irregularities are closely linked to the background electron density intensity. Xiong et al. (2016a) concluded that GPS signal reception may be interfered by

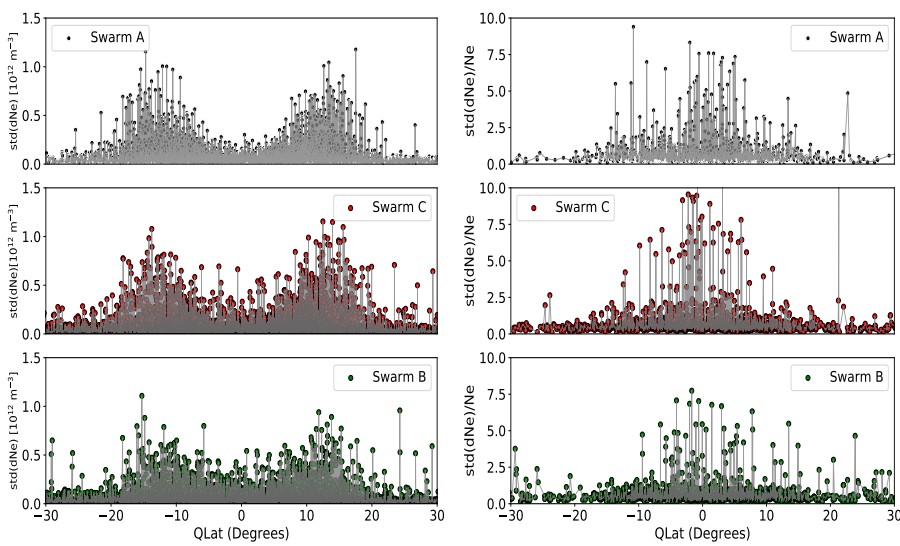

**Figure 8.** Scatter plots of latitude distribution of ionospheric irregularities observed for the period from October 2014 to October 2018. The left panel shows the distribution when irregularities are quantified using $std(dN_e)$, while the right panel shows the distribution when $std(dN_e)/N_e$ is used.

small-scale plasma density structures with large-density gradients in zonal and meridional directions. Here, we attempt to compare the seasonal and longitudinal distribution of electron density gradient in the meridional direction along the tracks of the Swarm satellites with the magnitudes presented in Fig. 6 and Fig. 7. To determine the $N_e$ gradient along the satellite tracks, $N_e$ depletion was divided by the corresponding latitudinal distance in degrees. Figure 9 presents the $N_e$ gradient, $\nabla N_e$ classified in different seasons for Swarm A, C, and B independently. The seasonal and longitudinal distribution of $\nabla N_e$ generally shows the same pattern as that of $std(dNe)$ and $std(dN_e)/N_e$ with high values observed during the equinoxes and December solstice and moderate values in the African sector in June solstice. However, close inspection of Fig. 9 shows that the $\nabla N_e$ has the same latitudinal distribution as $std(dN_e)$ i.e., it is symmetrical about the magnetic equator with high values at the EIA belts. On the other hand, the latitudinal distribution of $\nabla N_e$ is different from that of $std(dN_e)/N_e$ (see Fig. 7). Earlier studies have also shown that irregularity events at latitudes of the EIA might be associated with the regions of strong density gradient (e.g, Basu et al., 2001; Keskinen et al., 2003; Muella et al., 2008). The formation of small-scale irregularities appears to be more likely in ionospheric regions with higher background electron density and steep electron density gradients (Keskinen et al., 2003; Muella et al., 2008; Muella et al., 2010). Therefore, the amplitudes of ionospheric irregularities closely depend on background electron density (Wan et al., 2018) and steep $N_e$ gradient globally as expected.

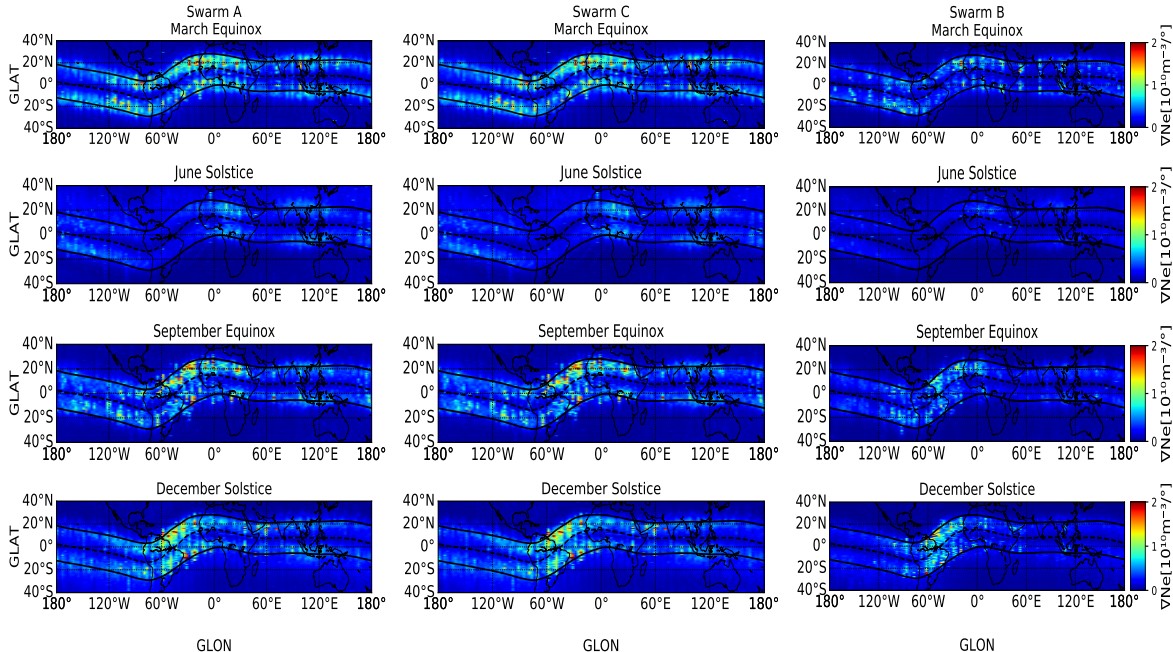

**Figure 9.** Along-track electron density gradient $\nabla N_e$ as derived from the Swarm satellites separated into four seasons (March/September equinox and June/December solstice) from October 2014 to October 2018 for Swarm A, C, and B. The black dotted line represents the magnetic equator, while the black solid lines represent the EIA belts at about $\pm 15°$ magnetic latitude.

### 3.3 Magnetic and Solar Activity Dependence of Ionospheric Irregularities

The Swarm faceplate observations began near solar maximum in October 2014 and approached solar minimum of solar cycle 24 towards the end of 2018. Figure 10 shows the Kp index and 10.7 cm solar radio flux (F10.7) index in units of $10^{-22}$ Wm$^{-2}$Hz$^{-1}$ to summarize the magnetic and solar activity for the period $2014 - 2018$. In general, solar cycle 24 was characterized by very low solar activity compared to cycles that preceded it (Basu, 2013). The F10.7 varied often between about 50 sfu and 200 sfu for period $2014 - 2018$. This period was also characterized by geomagnetic storms with Kp > 3. The effects of geomagnetic disturbances and changes in solar activity on ionospheric irregularity characteristics are of scientific interest and have been investigated in multiple studies (e.g, Palmroth et al., 2000; Sobral et al., 2002; Huang et al., 2002; Gentile et al., 2006; Stolle et al., 2006; Nishioka et al., 2008; Li et al., 2009; Basu et al., 2010; Sun et al., 2012; Carter et al., 2013; Huang et al., 2014). By using different criteria, Huang et al. (2014) determined the solar activity dependence of the occurrence of irregularities. In addition to solar activity, we also used different criteria to check the effects of magnetic variability on the distribution characteristics of irregularities in low latitudes.

Scatter plots of (a) std($dN_e$) and (b) std($dN_e$)/$N_e$ as functions of F10.7 are shown in Fig. 11 for Swarm A, B, and C, independently. To check on the solar activity dependence of std($dN_e$) and std($dN_e$)/$N_e$ at the equator and the EIA belts,

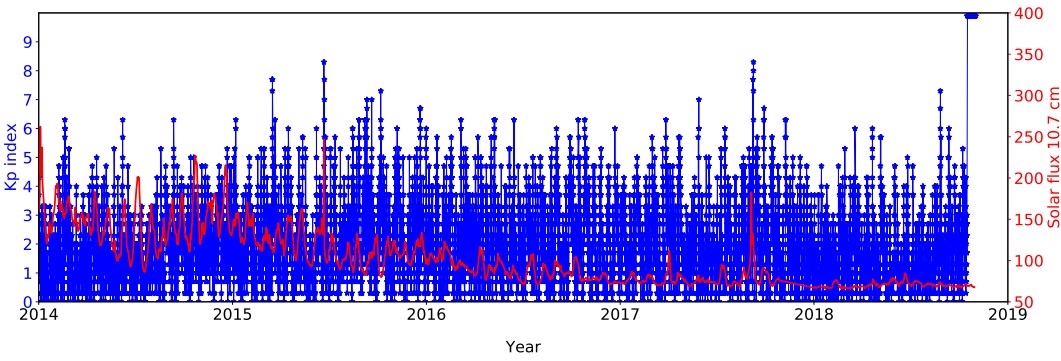

**Figure 10.** The 10.7 cm solar radio flux in solar flux units and the Kp index during $2014 - 2018$.

the Swarm satellite passes were divided into three latitudinal ranges i.e, Equator ($\pm 5°$ quasi-dipole latitude), Southern EIA region (-30°$-$ -5° quasi-dipole latitude), and Northern EIA region (+5°$-$+30° quasi-dipole latitude) (see legend of Fig. 11). Each panel of Fig. 11 contains linear fits and the correlation coefficients $R$. In general, both std($dN_e$) and std($dN_e$)/$N_e$ show weak positive correlation with F10.7 irrespective of the latitudinal range and this may be attributed to the small data-set used.

However, it can be seen that the correlation between F10.7 and std($dN_e$) is higher at the EIA regions than at the equator. This also shows the symmetrical distribution of std($dN_e$) with high values obtained at the EIA belts than at the equator. There is hardly any difference observed between equatorial and off equatorial latitudes for the case of std($dN_e$)/$N_e$ . The results obtained for std($dN_e$) is consistent with that of Liu et al. (2007) who presented the solar activity dependence of the electron density at the equatorial anomaly regions.

For Swarm A alone, Fig. 12 shows the solar variation effect on seasonal and longitudinal distribution of std($dN_e$) and std($dN_e$)/$N_e$ . The results are divided into two major columns (distribution with respect to std($dN_e$) to the left and std($dN_e$)/$N_e$ to the right). In each major column, there are two sub-columns, one for low solar activity ($F10.7 < 140$) and the other for moderate solar activity ($140 \leqq F10.7 < 180$). It is important to point out that a reduced number of days were used to generate the climatology maps when $140 \leqq$ F10.7 $< 180$ compared to when F10.7 $< 140$. In Fig. 12, high std($dN_e$) values are often

observed when $140 \leqq$ F10.7 $< 180$. On the contrary, high values of std($dN_e$)/$N_e$ are mostly observed when $F10.7 < 140$. The $F10.7$ dependence obtained using std($dN_e$) is similar to the results presented by Huang et al. (2001), Su et al. (2006), Stolle et al. (2006). It is necessary to note that Huang et al. (2001), Su et al. (2006), Stolle et al. (2006) addressed the solar activity dependence of the occurrence rate of ionospheric irregularities. Wan et al. (2018) presented differences between the occurrence rate of ionospheric irregularities and their amplitudes in terms of the latitudinal and longitudinal distribution. How-

ever, in general, the seasonal and longitudinal distribution of std($dN_e$) presented in Fig. 12 shows a similar dependence on different levels of F10.7 as the occurrence rates. Using simulations from the Magnetosphere - Thermosphere - Ionosphere - Electrodynamics General Circulation Model (MTIEGCM), Vichare and Richmond (2005) showed that upward evening drift increases at a similar rate in all longitude sectors with solar activity. Therefore, the high occurrence of irregularities during

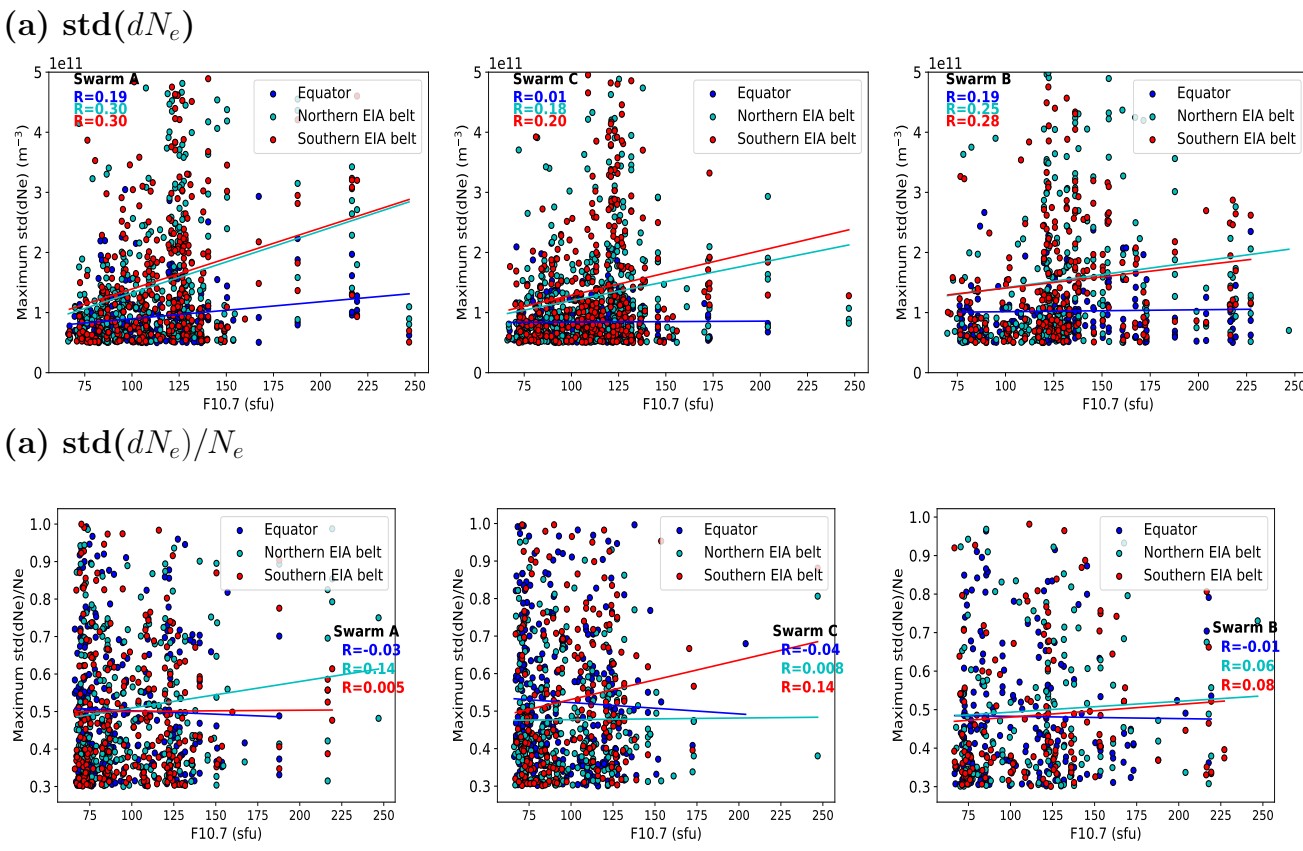

**Figure 11.** The distribution characteristics of (a) std($dN_e$) and (b) std($dN_e$)/$N_e$ with respect to 10.7 cm solar radio flux in solar flux units for the period from October 2014 to October 2018. The coloured solid lines in each panel represent a linear fit to the data.

moderate or high solar activity period may be because of the atmospheric driver for the zonal electric field which intensifies during moderate/high solar activity, causing an increase in the RTI growth rate.

Figure 13 presents scatter plots of (a) std($dN_e$) and (b) std($dN_e$)/$N_e$ as functions of Kp. To generate Fig. 13, the Swarm passes were also split into equatorial and EIA latitudes, similar to Fig. 11. In general, the results show a weak correlation
5  with Kp close to zero, irrespective of the method used to quantify the level of equatorial ionospheric irregularities and the latitude range. Close inspection of Fig. 13 shows that the correlation between std($dN_e$) and Kp was lowest at the EIA belts compared to that at the equator. There is hardly any difference observed between equatorial and off equatorial latitudes for the case of std($dN_e$)/$N_e$. Dao et al. (2011) adopted the relative perturbation to quantify irregularities. Their reason for using the relative perturbation was that the absolute perturbation is correlated with the ambient ion density, which varies due to
10  several factors such as varying altitude. The results shown in Fig. 11 and 13 also show that std($dN_e$) is more sensitive to solar and magnetic variations compared to std($dN_e$)/$N_e$. The differences in the correlation between F10.7 or Kp and std($dN_e$) or

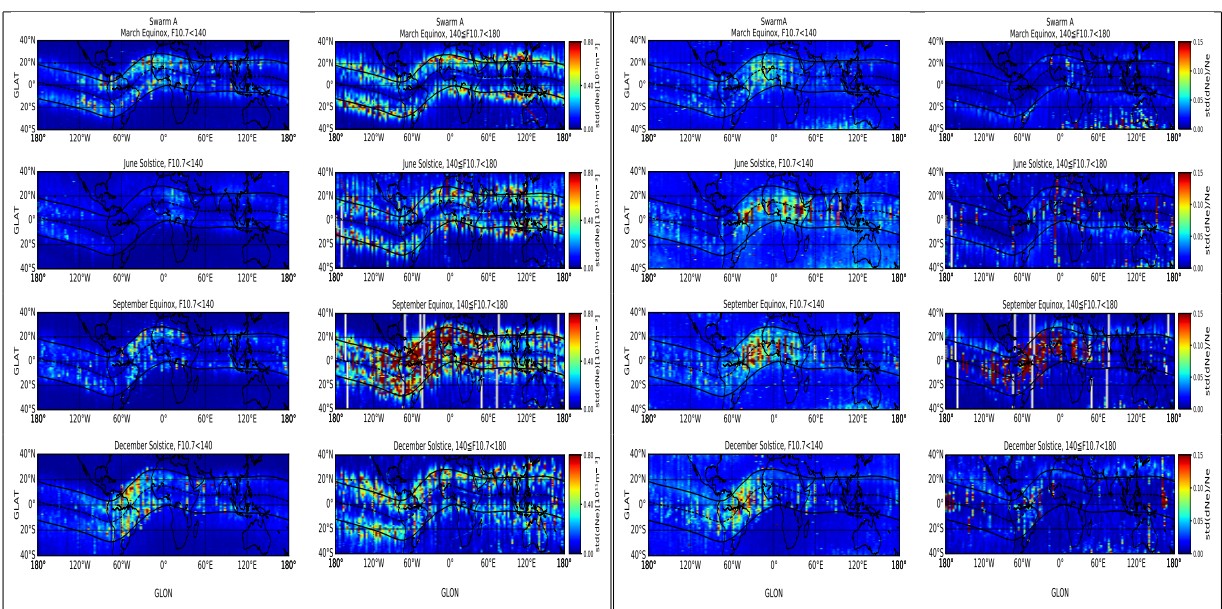

**Figure 12.** Solar activity effect on seasonal and longitudinal distribution of $\mathrm{std}(dN_e)$ and $\mathrm{std}(dN_e)/N_e$ for the period from October 2014 to October 2018: A case for Swarm A.

$\mathrm{std}(dN_e)/N_e$ at equatorial and off-equatorial latitudes can be explained by the differences in background electron density and electron density gradients at the crests and trough. Using DMSP pre-midnight plasma data, Huang et al. (2001) found that the rate of occurrence of irregularity and geomagnetic activity were negatively correlated. We also examined the geomagnetic effect on the seasonal and longitudinal distribution of irregularities as presented in Fig. 14 for Swarm A. The results are divided

5   into two major columns (distribution with respect to $\mathrm{std}(dN_e)$ to the left and $\mathrm{std}(dN_e)/N_e$ to the right). In each major column, there are two sub-segments, one for calm geomagnetic occasions (Kp < 3) and the other for geomagnetically disturbed periods (Kp $\geqq$ 3). From Fig. 14, high values of both $\mathrm{std}(dN_e)$ and $\mathrm{std}(dN_e)/N_e$ are frequently observed when Kp < 3. Palmroth et al. (2000) found a negative correlation between Kp and pre-midnight plasma depletions and a positive post-midnight correlation. They linked the distinction before and after local midnight to disturbed westward and eastward electrical fields, respectively.

10  Stolle et al. (2006) checked the response of the occurrence of ionospheric irregularities to geomagnetic activity using magnetic field measurements made by CHAMP and they observed a weak relation between the occurrence of irregularities and the Kp index. Huang et al. (2001) also used DMSP measurements of plasma density and found that the occurrence rate of ionospheric irregularities for low Kp values were almost doubled compared to when Kp values are high. The geomagnetic activity affects irregularity occurrence in the low latitudes in two noteworthy ways i.e., by the brief entrance of auroral electric fields (Fejer,

**(a)** $\mathbf{std}(dN_e)$

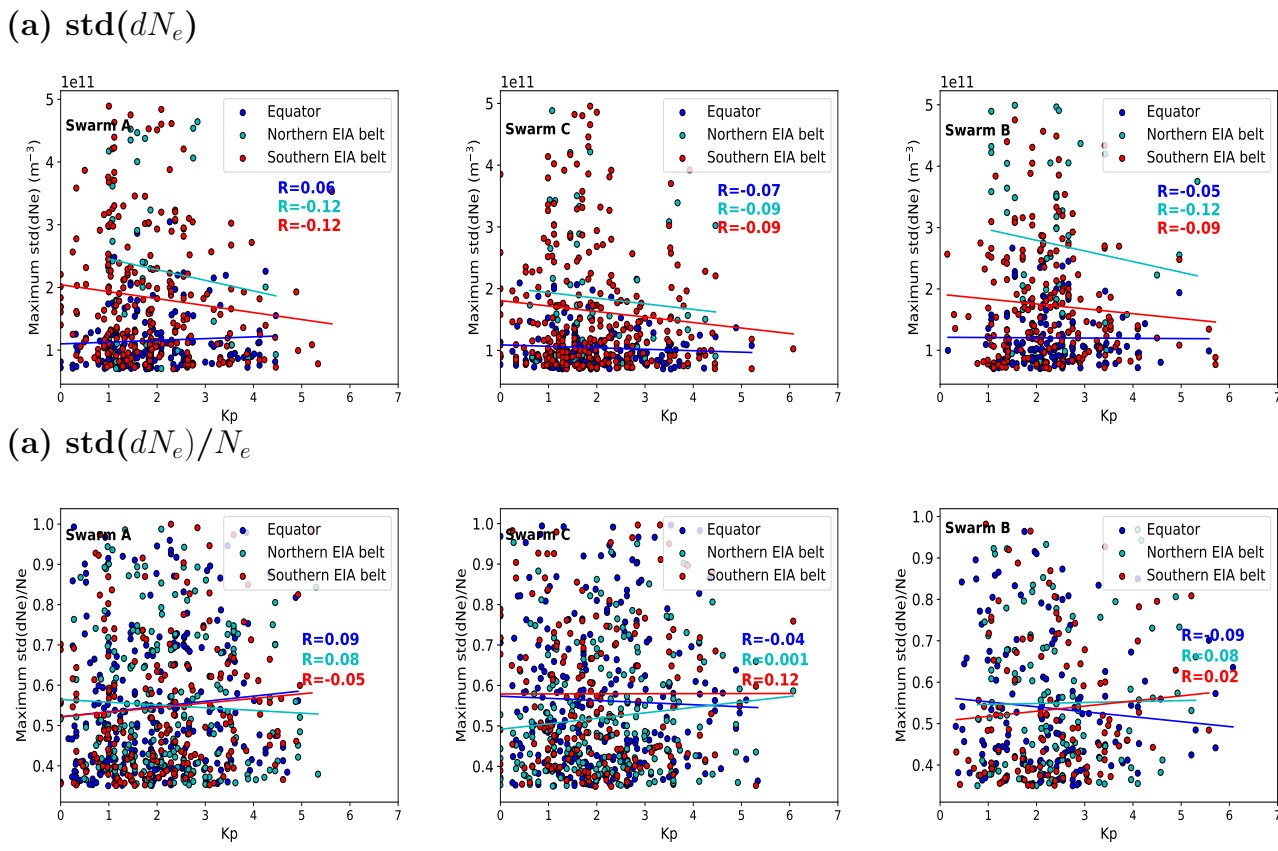

**(a)** $\mathbf{std}(dN_e)/N_e$

**Figure 13.** The distribution characteristics of (a) $\mathrm{std}(dN_e)$ and (a) $\mathrm{std}(dN_e)/N_e$ with respect to Kp index for the period from October 2014 to October 2018. The coloured solid lines in each panel represent a linear fit to the data.

1991; Kikuchi et al., 1996) and by the unsettling influence of dynamo effects (Blanc and Richmond, 1980). The second mechanism produces disturbance electric fields which last for a long time. The disturbance electric fields are westward after sunset (Blanc and Richmond, 1980; Huang et al., 2005; Abdu, 2012). It is important to note that the well-known trend in the longitudinal distribution of $\mathrm{std}(dN_e)$ and $\mathrm{std}(dN_e)/N_e$ for some seasons may not be clearly observed in Fig. 12 and Fig. 14
5 because of limited data after categorizing with respect to Kp or $F10.7$.

## 4  Conclusions

In this study, we have used Swarm $N_e$ data measured by the faceplate at a frequency of 16 Hz to examine the distribution characteristics of ionospheric irregularities in the equatorial and low latitude ionosphere from $2014-2018$ when the 16 Hz data was available. Two methods (absolute and relative perturbation) were used to quantify the level of ionospheric irregularities.

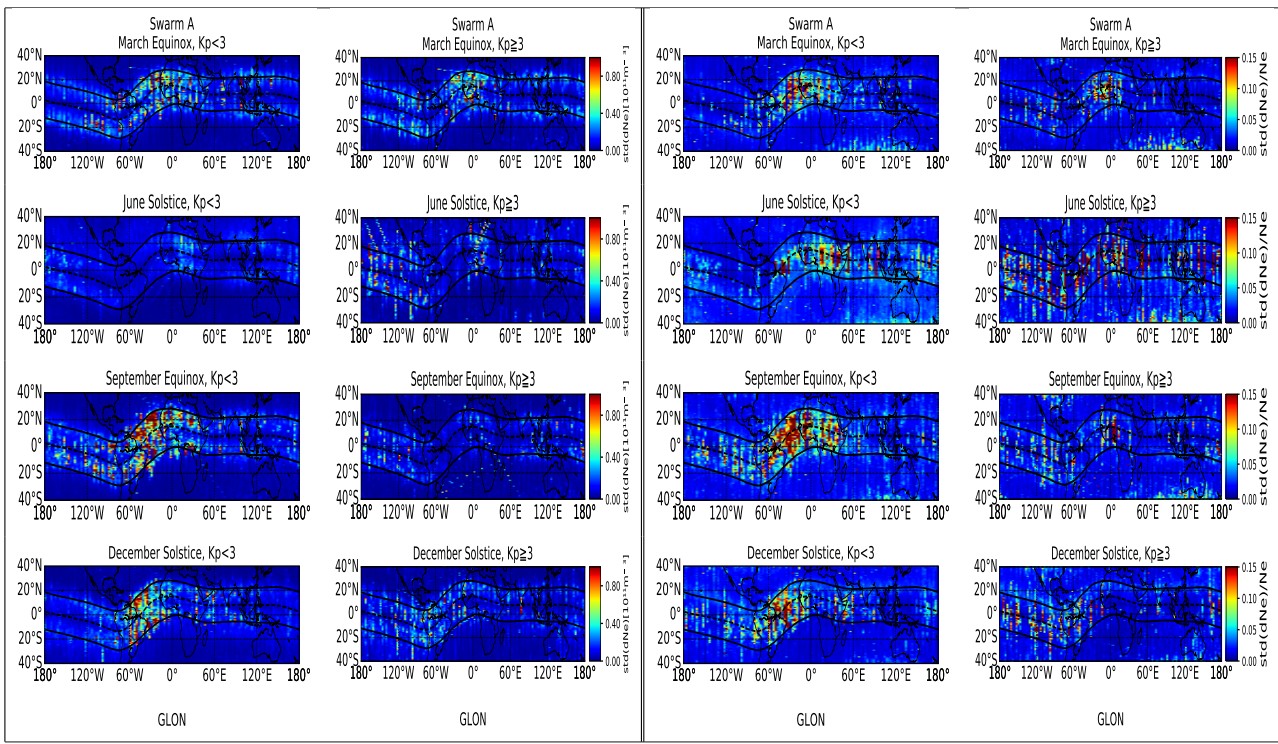

**Figure 14.** Geomagnetic effect on seasonal and longitudinal distribution of $\mathrm{std}(dN_e)$ and $\mathrm{std}(dN_e)/N_e$ for the period from October 2014 to October 2018: A case for Swarm A.

Both methods were able to capture fluctuations in electron density along single satellite passes. Basing on the large number of Swarm low latitude crossings for the years $2014 - 2018$, the local time, seasonal and longitudinal distribution of ionospheric irregularities in the low latitudes were examined. We demonstrated the importance of steep density gradients for the generation and distribution of ionospheric irregularities in the low latitudes. We also checked the effects of geomagnetic and solar activity

5 on the distribution characteristics of ionospheric irregularities. The findings are summarized below:

(i) The local time distribution of irregularities quantified by the two methods showed that they are mainly nighttime phenomena as expected. Both $\mathrm{std}(dN_e)$ and $\mathrm{std}(dN_e)/N_e$ showed similar trends in percentage occurrence during the equinoxes and December solstice with a peak occurrence between 2000 LT and 2200 LT. The percentage occurrence was lowest in June solstice. Generally, the local time dependence of ionospheric irregularities is not much different when either

10 $\mathrm{std}(dN_e)$ or $\mathrm{std}(dN_e)/N_e$ is used. However, the local time distribution according to $\mathrm{std}(dN_e)$ is closely related to that of $ROTI$ derived from ground-based stations.

(ii) In general, the seasonal and longitudinal distribution of ionospheric irregularities as quantified by the Swarm 16 Hz $N_e$ data was in agreement with past observations using other satellite missions irrespective of the method used. However,

close inspection of the magnetic latitudes reveals that $\text{std}(dN_e)$ and $\text{std}(dN_e)/N_e$ showed different latitudinal distribution of ionospheric irregularities about the magnetic equator. The $\text{std}(dN_e)$ showed a symmetric distribution about the magnetic equator with high magnitudes at latitudes of about $\pm 10° - \pm 15°$. The $\text{std}(dN_e)/N_e$ showed a peak at the quasi-dipole equator which gradually decreased towards higher latitudes.

(iii) The seasonal and longitudinal distribution of electron density gradient was closely related to that of $\text{std}(dN_e)$ and $\text{std}(dN_e)/N_e$. Also, symmetry about the magnetic equator was observed with $\nabla N_e$. Therefore, in addition to the background electron density presented by Wan et al. (2018), the longitudinal distribution of ionospheric irregularities also depends on steep electron density gradients as expected.

    (iv) The $\text{std}(dN_e)$ showed a weak positive correlation with $F10.7$ and the correlation was even lower with $\text{std}(dN_e)/N_e$.
Furthermore, $\text{std}(dN_e)$ in the EIA crest regions grew approximately linearly from the low to moderate solar activity, with higher correlation than that in the EIA trough region. The F10.7 dependence of the seasonal and longitudinal distribution of the ionospheric irregularities showed slightly different trends between $\text{std}(dN_e)$ and $\text{std}(dN_e)/N_e$. The discrepancy between the two methods may be because of the limited data. In general, the distribution of ionospheric irregularities was still lower during the geomagnetically disturbed period than in quiet times. The correlation between $\text{std}(dN_e)$ and Kp
was lowest at the EIA belts compared to that at the equator. The solar and magnetic activity dependence of $\text{std}(dN_e)/N_e$ hardly showed any difference in correlation between equatorial and off-equatorial latitudes.

Despite the obvious limitations of using polar-orbiting satellites to monitor equatorial electrodynamics, Swarm has provided credible distribution characteristics of ionospheric irregularities in the low latitude region with data accumulated in five years $(2014 - 2018)$. In general, the initial observations of the distribution characteristics of ionospheric irregularities using the 16
20   Hz $N_e$ data are in good agreement with earlier works that have addressed similar concepts. This has demonstrated the ability of Swarm faceplate $N_e$ data for ionospheric studies. Therefore, the 16 Hz faceplate data is a useful measurement that can be adopted in order to understand ionospheric irregularities.

*Data availability.* The official website of Swarm is `http://earth.esa.int/swarm` and `ftp://swarm-diss.eo.esa.int` is the server for the distribution of Swarm data. IGS/GPS data for the equatorial and low latitude stations were downloaded from
`http://garner.ucsd.edu/pub/rinex/`. The Kp index and F10.7 solar radio flux data used in this study were obtained from the website `http://omniweb.gsfc.nasa.gov/`.

*Author contributions.* Aol Sharon, Stephan Buchert, and Edward Jurua designed the concepts and implemented them. Aol Sharon prepared the manuscript with contributions from all co-authors.

*Competing interests.* No competing interests are present.

*Acknowledgements.* This study was sponsored by the International Science Program (ISP) of the Uppsala University, Sweden. The authors acknowledge ESA Swarm team for the Swarm mission and for providing the Swarm data. The official Swarm website is `http://earth.esa.int/swarm`, and the server for Swarm data distribution is `ftp://swarm-diss.eo.esa.int`. IGS/GPS data for the equatorial and low latitude stations were downloaded from `http://garner.ucsd.edu/pub/rinex/`. The Kp index and F10.7 solar radio flux data used in this study were obtained from the website `http://omniweb.gsfc.nasa.gov/`.

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
