# Peer review of "Traits of sub-kilometer F-region irregularities as seen with the Swarm satellites"

_Annales Geophysicae, 2019_

## Referee Comment (RC1) · Anonymous Referee #1 · 4 May 2019

The manuscript "Traits of sub-kilometer F-region irregularities as seen with the Swarm satellites" by Aol et al. reviews post-sunset equatorial fluctuations identified in electron density time series obtained by the TII faceplate technique onboard the Swarm satellites with 16Hz sampling rate. The results are distributions of the occurrence rate of the identified fluctuations over local time separately for 4 seasons (not separated by longitude), and of the amplitude of the identified fluctuations over longitude and latitude separately for 4 seasons (not separated in local time). The authors further conclude that the solar flux and geomagnetic activity and these amplitudes are weakly correlated.

General comments:

The results shown confirm earlier published findings, and some other analyses do not

seam fully consistent (see specific comments). The submitted manuscript provides an excellent report to indicate that identified fluctuations in electron density time series obtained by the TII faceplate technique onboard the Swarm satellites with 16Hz sampling rate are most probably related to equatorial post-sunset plasma irregularities. But it is unlikely that the report enhances present scientific knowledge and evidence.

My major concern is, however, that this analysis does not show any findings based on the 16Hz samplings that could not have been achieved with the 2Hz samplings onboard Swarm. The authors are encouraged to exploit the high value of the 16Hz sampling.

Selected specific comments:

The concept of deriving electron density from the TII faceplate technique is an extended product in the concept of the Swarm mission. A bit a more detailed description of the retrieval and the data shall be added, next to refering to "Buchert, S.: Extended EFI LP data FP release notes, ESA Technical Note, 2016." (I did not find the document in the web, and do not know if/where it is accessible.)

P2, L1-14: The authors have access to the full Swarm mission data to perform orbital analyses, such as the spacing of Swarm A/C or the local time processing. Alternatively, they may refer to classical papers, such as provided in doi:10.5047/eps.2013.07.001 if needed. You might reconsider, if Kil et al. or Xiong et al. are suitable references in this broad context.

P2, L25ff: The proposed detection method identifies amplitudes of deviations from a 2s-average and attribute a level of higher than 0.25*10ˆ10 mˆ-3 of its RMS as being an irregularity. By this method, identifying an RMS over a time window of 2 seconds (15km), information from the 16Hz seconds are smoothed out.

Figure1/P5, L3: 'multi-peak' variations appropriate to the 16Hz samples in comparison to the 2Hz samples cannot be identified in either the zoomed figures in Figure 1. Many peaks are equally visible in the 2Hz data.

Page 10,L9: "Large values of both ΔNe and ΔNe /Ne often occur in locations of large depletions in Ne." It is not clear, where else they could be expected?

Page 5, L15-23: The effect of relative variation compared to absolute variation was extensively discussed, e.g., for polar patches in DOI: 10.1002/2017JA024811 .

P7, L3: The authors describe that the irregularities occur between 18-06LT, however, no other LT is shown in Figure 3.

Figures 6,7,12 base on color scales that lie below the detection threshold by at least to 50%.

Figure 9 shows along track gradients of electron density, that are by nature directly related to ΔNe for the detections. It is maybe not helpful to discuss their coincidence.

Figure 12 divides below and above F10.7=140sfu. It is questionable if a significant amount of data is available for conclusions above F10.7=140sfu. See figures 10, 11 of the submission and https://services.swpc.noaa.gov/images/solar-cycle-10-cm-radio-flux.gif. Also, P 14 L 20ff: The authors mention similar comparisons by Huang et al., Su et al., Stolle et al. To the knowledge of the referee, these papers discussed the occurrence rate, not the amplitude of irregularities. Amplitudes are discussed by Wan et al., 2018. However, conclusion (iv) of the submitted paper mentions occurrence rates which is not compatible with Figure 12.

Palmroth et al., 2000 and Stolle et al., 2006 did first analyses on the relation between the occurrence rate of irregularities and the Kp index, that are not discussed in the submitted manuscript.

The authors might reconsider the added value of P11 L6ff to the concerned study.

Selected technical comment:

dNe, ΔNe, and (nabla)Ne are used to express the same parameter.

---

## Referee Comment (RC2) · Anonymous Referee #2 · 12 May 2019

Using the 16 Hz Ne data of Swarm satellites, authors identify the small-scale F-region irregularities which is not distinguishable from those of large-scale, according to their algorithm. Authors investigated the local time variation of the occurrence rate, the seasonal/longitudinal (s/l) variation of absolute electron density perturbation as well as that of the relative electron density perturbation ( $\Delta$ Ne/Ne), latitude distribution of  $\Delta$ Ne and  $\Delta$ Ne/Ne, s/l variation of Ne gradient, correlations between the Ne perturbations with the Solar Flux as well as Kp are also presented. The works validated the quality of the Swarm 16 Hz Ne data and its capability of characterizing the small-scale irregularities. Some of the finds are in consistent with previous works. However, from my perspective, few critical issues need to clarify and improved.

Major comments:
First of all, I would like to make some clarification according to my understanding of the manuscript. 1. The small-scale irregularities are usually embedded in the large-scale irregularities (seen Figure 2 and from my experience), and they are actually the regional density fluctuation inside the plasma bubbles (PBs). 2. According to the authors' definition on the  $\Delta$ Ne, that is, the standard deviation of the residuals between the original data with the mean fitting, in 32 data points (2 seconds, 0.6 degrees in latitudes, 14 Km in length). The  $\Delta$ Ne quantified the density fluctuation in a spatial scale of 0.6ËŽ GLAT (normally inside a major PB), which is different from the conception of the depletion amplitudes, in my opinion.

(1) The author should emphasize in the article that these small-scale irregularities is not independent and has not been distinguished, from those of large-scale (PBs). (2) Authors compared the occurrence of scintillations (quantified by ROTI) in 9 stations with the small-scale irregularities which exhibit correlations. However, as noted in previous works (e.g. Xiong et al., 2016; Wan et al., 2018), the radio signal disruption is more likely to occur when there exists PBs with large depletion amplitude. In the same reason as described in my first clarified issue, I think authors should be careful to relate the two things up. (3) P13 Line10 – 12: Obviously, the s/l variation of  $\hat{a}L\tilde{G}Ne$  is similar to  $\Delta Ne$  but not  $\Delta Ne/Ne$ , authors should describe the plots objectively.

Minor comments: (1) Figure 1: I would recommend the author to plot all the profiles using the absolute scale. (2) P8, L19 - 20: in pre-midnight sector. (3) P8, L20 – P9, L2: It's not appropriate to explain that in a decayed sense, since the occurrence of irregularities at post-midnight is enhanced as authors found and claimed, the mechanism should be related to the lower depletion amplitudes. (4) Figure 5b: It seems that the scintillation data is less available during June Solstice compared to other seasons for some stations. E.g. RIOP is totally missing. The author should mention those in the article. (5) Figure 11& Figure 13: I suggest the authors do the correlation analysis regards to different latitudinal bins (i.e. equator and off-equator). (6) P13 Line 12, Linking word is missing. (7) P17, Line 11-12: It is not an accurate description. (8) In many
places, authors inappropriately describe the  $\Delta Ne$  and  $\Delta Ne/Ne$  as the 'irregularities' (e.g. the captions of Figure 13, 14).

References: Wan, X., Xiong, C., Rodriguez-Zuluaga, J., Kervalishvili, G. N., Stolle, C., and Wang, H.: Climatology of the Occurrence Rate and Amplitudes of Local Time Distinguished Equatorial Plasma Depletions Observed by Swarm Satellite, Journal of Geophysical Research, Space Physics, 123, 3014–3026, https://doi.org/10.1002/2017JA025072, 2018 Xiong, C., Stolle, C., & Lühr, H. (2016). The Swarm satellite loss of GPS signal and its relation to ionospheric plasma irregularities. Space Weather, 14, 563–577. https://doi.org/10.1002/2016SW001439

---

## Author Comment (AC1) · 16 May 2019

**Response to Reviewer 1 comments**

The authors of this manuscript thank the reviewer for the suggestions and comments. Our response to the comments and suggestions are presented below.

**General Comments**

- The results shown confirm earlier published findings, and some other analyses do not seam fully consistent (see specific comments). The submitted manuscript provides an excellent report to indicate that identified fluctuations in electron density time series obtained by the TII faceplate technique onboard the Swarm satellites with 16Hz sampling rate are most probably related to equatorial post-sunset plasma irregularities. But it is unlikely that the report enhances present scientific knowledge and evidence.

  My major concern is, however, that this analysis does not show any findings based on the 16Hz samplings that could not have been achieved with the 2Hz samplings onboard Swarm. The authors are encouraged to exploit the high value of the 16Hz sampling.

- **Response:**

  - Thank you for your valuable feedback. Our research is the first to use the measurements of Swarm 16-Hz faceplate electron density to characterize ionospheric irregularities. This study also directly compares the relative and absolute perturbation of electron density from a meridional point of view using data from Swarm 16 Hz over low latitude.

  - This study aimed at checking the capability of the Swarm 16-Hz faceplate electron density measurements for ionospheric irregularity observations. In this study we do not disregard the capability of the 2 Hz electron density measurements made by the Langmuir Probes on board Swarm. However, as stated in P.2,L.28, high-resolution data enables smaller scale structures to be identified in electron density (Nishioka et al., 2011,https://doi.org/10.1029/2011ja016446).

  - We agree with the reviewer about exploiting further the 16 Hz Swarm electron density measurements. In fact, Alfonsi et al., 2007, http://dx.doi.org/10.1016/j.asr.2017.05.021 recommended Swarm high resolution electron density measurements as input to the WAM model. However, as stated in P.2, L.7-9, the dependence of ionospheric irregularities on the geophysical parameters, remains a problem in modeling their variation for predictive purposes (Yizengaw and Groves, 2018,https://doi.org/10.1029/2018sw001980). The results presented in this study, characterize ionospheric irregularities basing on various geophysical parameters as a first step towards developing a model from Swarm 16-Hz in situ measurements of electron density. Currently, the team involved in this paper are carrying out further research involving modeling of amplitude scintillation from Swarm 16 Hz measurements using the WAM model. Thank you for your suggestion.

**Selected specific comments:**

1. The concept of deriving electron density from the TII faceplate technique is an extended product in the concept of the Swarm mission. A bit a more detailed description of the retrieval and the data shall be added, next to refering to "Buchert, S.: Extended EFI LP data FP release notes, ESA Technical Note, 2016." (I did not find the document in the web, and do not know if/where it is accessible.)

- **Response:**

  - The ESA Technical report and a link to the 16 Hz electron density data are accessible to everyone via ftp at: `ftp://swarm-diss.eo.esa.int/Advanced/Plasma_Data/16_Hz_Faceplate_plasma_density`.

  - More detailed information about obtaining electron density from the TII can also be obtained from Knudsen et al, 2017, doi:10.1002/2016ja022571.

2. P2, L1-14: The authors have access to the full Swarm mission data to perform orbital analyses, such as the spacing of Swarm A/C or the local time processing. Alternatively, they may refer to classical papers, such as provided in doi:10.5047/eps.2013.07.001 if needed. You might reconsider, if Kil et al. or Xiong et al. are suitable references in this broad context.

- **Response:**

  – We agree with the reviewer that it would be interesting to perform an orbital analysis for the available Swarm 16 Hz data set. In fact, an earlier study presented by Xiong et al. (2016),https://doi.org/10.1186/s40623-016-0502-5, utilized the strategic orientation of Swarm A and C (i.e, spacing between the two satellites) to check the scale sizes of irregularity structures using Swarm 2 Hz electron density measurements. The current study was designed to focus mainly on characterizing ionospheric irregularities using the Swarm 16 Hz electron density measurements and we'd like to keep the focus intact. Thank you for the suggestion.

3. P2, L25ff: The proposed detection method identifies amplitudes of deviations from a 2s-average and attribute a level of higher than $0.25 \times 10^{10} \text{m}^{-3}$ of its RMS as being an irregularity. By this method, identifying an RMS over a time window of 2 seconds (15km), information from the 16Hz seconds are smoothed out.

- **Response:** The window to calculate dNe should be short, but has to be long enough to avoid spurious detection of irregularities. We also tried 1 s, 16 points, instead of 2s, 32 points, and the outcome was similar and reasonable. It seemed a good compromise to go for RMS at 2 s (15 km) to quantify ionospheric irregularities.

4. Figure1/P5, L3: 'multi-peak' variations appropriate to the 16Hz samples in comparison to the 2Hz samples cannot be identified in either the zoomed figures in Figure 1. Many peaks are equally visible in the 2Hz data.

- **Response:** We present here inset plots which are zoomed in further. It can now be clearly seen that multiple

electron density peaks are observed on the 16 Hz data than in the 2 Hz data. The 2Hz data appears to be smooth and cannot show detailed small scale structures.

5. Page 10,L9: "Large values of both $\Delta N_e$ and $\Delta N_e/N_e$ often occur in locations of large depletions in Ne." It is not clear, where else they could be expected?

- **Response:** In the low latitudes, the values of both parameters are expected to be high in locations of plasma density depletions between the EIA crests close to the geomagnetic equator, but also at the bottom of large scale bubbles/depletions. However, $\Delta N_e$ is high at the crests and at the edge of large scale bubbles/depletions. So high values of both $\Delta N_e$ and $\Delta N_e/N_e$, do not occur at the same place, but at different places. We have shown this for the crests/equator in Figures 6 and 7.

6. Page 5, L15-23: The effect of relative variation compared to absolute variation was extensively discussed, e.g., for polar patches in DOI: 10.1002/2017JA024811 .

- **Response:**We thank the author for citing out relevant literature. In the present manuscript we extended the comparison of relative and absolute variation in the low latitude region using the polar orbiting Swarm satellites, where the distribution of ionospheric irregularities seems to be quite different from that at the polar region.

7. P7, L3: The authors describe that the irregularities occur between 18-06 LT, however, no other LT is shown in Figure 3.

- **Response:**Indeed as noticed by the reviewer, other local times are not included in Figure 3. The local time sector was fixed between 1800-0600 LT since over the low latitude region, ionospheric irregularities are a night time phenomena and this has been stated in the manuscript on Page 5, L. 25-27. The reviewer is referred to the the literature cited in Page 5, L. 25-26. Most of the earlier studies cited in Page 5, L. 25-26 also limited their analysis over the low latitude region to the local time sector 1800-0600 LT.

8. Figures 6,7,12 base on color scales that lie below the detection threshold by at least to 50%

- **Response:** To generate climatology maps presented in Figures 6,7,12,14, all Swarm passes were considered irrespective of whether there were irregularities or not. This was done because of the limited data.

9. Figure 9 shows along track gradients of electron density, that are by nature directly related to $\Delta N_e$ for the detections. It is maybe not helpful to discuss their coincidence.

- **Response:** As the reviewer has mentioned, the along track gradients are indeed directly related to $\Delta N_e$. An observation of interest was that there was a slight latitudinal difference between $\Delta N_e/N_e$ and $\nabla N_e$ and therefore, it was important to include $\nabla N_e$ and compare it to $\Delta N_e/N_e$ and $\Delta N_e$. Also, from the best of our knowledge, seasonal dependence of latitudinal-longitudinal distribution of $\nabla N_e$ has not yet been presented for the case of the Swarm satellites.

10. Figure 12 divides below and above F10.7=140sfu. It is questionable if a significant amount of data is available for conclusions above F10.7=140sfu. See figures 10, 11 of the submission and https://services.swpc.noaa.gov/images/solar-cycle-10-cm-radio- flux.gif. Also, P 14 L 20ff: The authors mention similar comparisons by Huang et al., Su et al., Stolle et al. To the knowledge of the referee, these papers discussed the occurrence rate, not the amplitude of irregularities. Amplitudes are discussed by Wan et al., 2018. However, conclusion (iv) of the submitted paper mentions occurrence rates which is not compatible with Figure 12.

- **Response:**

  - For the case of Fig. 12, we agree with the reviewer that the data may not have been enough after categorizing with respect to $F10.7$ especially for $140 \leq F10.7 < 180$. This limitation was also stated in the manuscript in Page 14,L 19-20 and Page 16,L 1-2. However, irrespective of the limitation in data availability, $\Delta N_e$ shows dependence on moderate solar activity similar to what was presented in earlier studies i.e, high $\Delta N_e$ values are often observed when $140 \leq F10.7 < 180$.

  - As pointed out by the referee, it is true that Huang et al. (2001), Su et al. (2006), Stolle et al. (2006) did not present amplitudes of electron density perturbation, and amplitudes are discussed by Wan et al. 2018. However, Wan et al. 2018 did not present the dependence of amplitudes of ionospheric irregularities to different solar activity levels categorized in terms of $F10.7$, while the cited papers (Huang et al. (2001), Su et al. (2006), Stolle et al. (2006)) addressed the solar activity dependence of occurrence of ionospheric irregularities. A key difference between occurrence of ionospheric irregularities and their amplitudes stated by Wan et al. 2018 is that they show a totally different longitudinal pattern. The amplitudes presented in Fig. 12 seem to show similar dependence on different levels of $F10.7$ as the occurrence rates presented by Huang et al. (2001), Su et al. (2006), Stolle et al. (2006) with high $\Delta N_e$ values (or high occurrence rates) often observed when $140 \leqq F10.7 < 180$. Therefore, the dependence of amplitudes, $\Delta N_e$ and occurrence rate of ionospheric ionospheric show similar dependence on solar activity level.

  - Conclusion (iv) will be rephrased in terms of the amplitudes.

11. Palmroth et al., 2000 and Stolle et al., 2006 did first analyses on the relation between the occurrence rate of irregularities and the Kp index, that are not discussed in the submitted manuscript.

- **Response:** We thank the reviewer for citing relevant literature.

12. The authors might reconsider the added value of P11 L6ff to the concerned study.

- **Response:** Thank you for the suggestion. As stated in response to comment 2., orbital analysis such as spacing between Swarm A and C and altitude difference between Swarm A/C and B etc is interesting. In this paper we prefer to focus on traits of sub kilometer F region ionospheric irregularities, and so we will leave the orbital analysis aspect for further research.

  **Selected technical comment:** dNe, $\Delta N_e$, and (nabla)Ne are used to express the same parameter.

- **Response:**

  - In the revised manuscript, we will replace $\Delta N_e$ with std($dN_e$) to represent the absolute electron density perturbation and it is obtained by determining the standard deviation of the residual $dN_e = N_e - \overline{N_e}$.

  - $\nabla N_e$ in this manuscript represents the electron density gradient derived along the satellite tracks and it is given by: $\nabla N_e = \frac{(N_e)_f - (N_e)_i}{X_f - X_i}$, where $X$ in this formula represents the latitude, $f$ is the final position and $i$ is the initial position.

  **Final Remark :** We thank the reviewer for the multiple comments.

---

## Referee Comment (RC3) · Anonymous Referee #3 · 27 May 2019

In this paper, the ionospheric irregulariteis are studied by using two methods: the absolute and relative changes of electron density. The distribution characteristics with local time, season and longitude of irregular structures are reproduced. The differences in the latitudinal distribution, and the correlation difference with F107 between the two methods are reported.

The main problem for the reviewer is that the relative change depends on both the absolute change in electron density and the value of background electron density. The strong relative change in the equatorial region may be due to the weaker background electron density. The dependence on F107 is not strong and may also be due to changes in background electron density with F107. From this point of view, perhaps absolute changes may be more suitable for studying the ionospheric irregulariteis. So

I don't know why the author should use the relative change as a way to study the ionospheric irregulariteis? I hope they can explain why.

More interesting is the finding of the importance of the meridional difference in electron density in the irregular structure of the ionosphere. Could they explain more about the physical mechanism?

---

## Author Comment (AC2) · 4 Jun 2019

The comment was uploaded in the form of a supplement:
https://www.ann-geophys-discuss.net/angeo-2019-50/angeo-2019-50-AC2-supplement.pdf

---

## Author Comment (AC4) · 6 Jun 2019

**Response to Reviewer 2 comments**

The authors of this manuscript thank the reviewer for the suggestions. Our response to the comments and suggestions are presented below.

First of all, I would like to make some clarification according to my understanding of the manuscript. 1. The small-scale irregularities are usually embedded in the large- scale irregularities (seen Figure 2 and from my experience), and they are actually the regional density fluctuation inside the plasma bubbles (PBs). 2. According to the authors' definition on the $\Delta N_e$, that is, the standard deviation of the residuals between the original data with the mean fitting, in 32 data points (2 seconds, 0.6 degrees in lat- itudes, 14 Km in length). The $\Delta N_e$ quantified the density fluctuation in a spatial scale of 0.6° GLAT (normally inside a major PB), which is different from the conception of the depletion amplitudes, in my opinion.

**Response:**(1.) We agree with the reviewer and we are aware of this fact.(2.) We also agree that the concept of a depletion amplitude is less applicable for the small-scale amplitudes analyzed in this work, and the submitted text is not clear about this. The revision will be improved in this respect. As a new aspect which is possibly relevant for especially the small-scale irregularities we have analyzed the along-track gradient of Ne (as a proxy of the pressure gradient).

**Major Comments**

(1) The author should emphasize in the article that these small-scale irregularities is not independent and has not been distinguished, from those of large-scale (PBs).

**Response:** We thank the reviewer for the suggestion and this will be emphasized in the revised manuscript.

(2) Authors compared the occurrence of scintillations (quantified by ROTI) in 9 stations with the small-scale irregularities which exhibit correlations. However, as noted in previous works (e.g. Xiong et al., 2016; Wan et al., 2018), the radio signal disruption is more likely to occur when there exists PBs with large depletion amplitude. In the same reason as described in my first clarified issue, I think authors should be careful to relate the two things up.

**Response:**We appreciate the reviewer for the valuable comment. We are aware of the fact that radio signal signal disruption is more likely to occur when there exists PBs with large depletions. We also sated in Page 2, L 12-13 that "an interesting feature of these plasma irregularities is their scale sizes. They typically cover a variety of scale sizes, from a few centimeters to thousands of kilometers (Zargham and Seyler, 1989 , https://doi.org/10.1029/ja094ia07p09009 ; Hysell and Seyler, 1998,https://doi.org/10.1029/98ja02616)". The bubbles themselves have horizontal sizes of order 100 km. The 9 IGS stations used in this study transmit radio signals at two L-band frequencies i.e, L1 (1575.42 MHz) and L2 (1227.60 MHz). The small-scale electron density variations commonly termed as small-scale structures are known to affect more L-band transmissions (Luhr et. al. 2014, doi: 10.3389/fphy.2014.00015, Bhattacharyya et. al. 2003, doi:10.1029/2002RS002711, Rao et. al. 2005, doi:10.1007 / s00585-997-0729-3, Spogli et. al. 2016, doi:10.1002/2016JA023222). The Swarm high resolution 16 Hz density estimates correspond to a spatial resolution of about 500 m, which is within the range of applicable Fresnel scales, and so theoretically relevant as a cause of L-band scintillations. Therefore, in the current manuscript, we compared small scale irregularity structures quantified from the Swarm 16 Hz electron density measurements with ROTI derived from VTEC measurements at L-band frequency.

(3) P13 Line10 − 12: Obviously, the s/l variation of $\nabla N_e$ is similar to $\Delta N_e$ but not $\Delta N_e/N_e$, authors should describe the plots objectively.

**Response:** The seasonal and longitudinal distribution of $\nabla N_e$ generally shows the same pattern as that of $\Delta N_e$ and $\Delta N_e/N_e$ with high values observed during the equinoxes and December solstice and moderate values in the African sector in June solstice. However, close inspection of Figure 9 shows that the $\nabla N_e$ has the same latitudinal distribution as $\Delta N_e$ i.e., it is symmetrical about the magnetic equator with high values at the

EIA belts. On the other hand, the latitudinal distribution of $\nabla N_e$ is different from that of $\Delta N_e/N_e$. Earlier studies have also shown that irregularity events at latitudes of the EIA might be associated with the regions of strong density gradient (e.g, Basu et al., 2001,https://doi.org/10.1029/2001ja001116; Keskinen et al., 2003,https://doi.org/10.1029/2003gl017418). The formation of small-scale irregularities appears to be more likely in ionospheric regions with higher background electron density and steep electron density gradients (Keskinen et al. 2003,https://doi.org/10.1029/2003gl017418; Muella et al. 2008,https://doi.org/10.1029/2007ja012605; Muella et al. 2010,https://doi.org/10.1029/2009JA014788). Therefore, the amplitudes of ionospheric irregularities closely depend on background electron density (Wan et al. 2018) and steep $N_e$ gradient globally, as expected. We shall describe the plots objectively in the revised manuscript.

**Minor comments**:

(1) Figure 1: I would recommend the author to plot all the profiles using the absolute scale.

**Response:** We thank the author for the recommendation. However, the trend is similar whether the absolute or logarithmic scale is adopted for Figure 1. We present here a comparison for Swarm A on 2014-10-06.

[Figure]

[Figure]

(2) P8, L19 - 20: in pre-midnight sector.

**Response:** In P8, L20-P9, L2, we were describing the enhanced post-midnight irregularities seen in Fig. 2(b) for June Solstice. The sentence will be rephrased to make it more clear.

(3) P8, L20 – P9, L2: It's not appropriate to explain that in a decayed sense, since the occurrence of irregularities at post-midnight is enhanced as authors found and claimed, the mechanism should be related to the lower depletion amplitudes.

**Response:** We do not claim that the enhanced irregularities post-midnight is related to lower depletions. The mechanisms that generate these post-midnight irregularities are still unknown and widely debated. Two mechanisms were suggested to explain the occurrence of post-midnight irregularity. One mechanism is the seeding of the RTI by atmospheric gravity waves originating from below into the ionosphere, while the other mechanism is the elevation of the F-layer by the meridional neutral winds in the thermosphere, which may be associated with the maximum midnight temperature in the thermosphere (Otsuka, 2018, https://doi.org/10.1186/s40645-018-0212-7, and references therein). The sentence will be rephrased to make it clear.

(4) Figure 5b: It seems that the scintillation data is less available during June Solstice compared to other seasons for some stations. E.g. RIOP is totally missing. The author should mention those in the article.

**Response:**We thank the reviewer for the observation. The IGS station RIOP did not have data in June solstice and September equinox. This information will be included in the revised manuscript.

(5) Figure 11& Figure 13: I suggest the authors do the correlation analysis regards to different latitudinal bins (i.e. equator and off-equator).

**Response:**We agree with the reviewer that the geomagnetic and solar activity dependence of occurrence of ionospheric irregularities at different latitudinal regions (i.e, equator and off-equator) would be interesting to know. In fact, Liu et. al. 2007, JGR,112, A11311, doi:10.1029/2007JA012616 investigated the solar activity dependence of the electron density at equatorial and low latitudes. We present here a snapshot of one of the results extracted from Liu et. al. 2007 concerning latitudinal distribution on solar activity dependence of electron density. From the

[Figure]

**Figure 8.** Same as Figure 7 but for postsunset (1800–2300 MLT) sector for the (left) northern EIA crest, (middle) dip equator, and (right) southern EIA crest.

results presented by Liu et. al. 2007, the electron density in the crest regions of the EIA grows roughly linearly from solar minimum to solar maximum, with higher growth rate than in the EIA trough region. However, Liu et. al. 2007 did not quantify ionospheric irregularities in their analysis. Therefore, we attempted the suggestion and the result is presented in Figure 1 for Swarm A. We divided the Swarm satellite passes into three latitudinal ranges i.e, Equator ($\pm 5°$ quasi-dipole latitude), Southern EIA region (-30°− -5° quasi-dipole latitude), and Northern EIA region (+5°−+30° quasi-dipole latitude). A summary of the latitudinal ranges is presented in a map in Figure 2. The correlation is generally low irrespective of the latitudinal region and this may be attributed to the small data sample. However, it can be seen that the correlation between F10.7 and std($dN_e$) is higher at the EIA regions than at the equator. Similar observation was made for Swarm C and B (not presented here). This shows also the symmetrical distbution of std($dN_e$) with high values obtained at the EIA belts than at the equator as seen in Fig. 6. For the case of std($dN_e$) dependence on Kp, it is a generally weak correlation and it is more negative at the equator than off equatorial latitudes. For std(dNe)/dNe, the correlation is very weak with both kp and F10.7. There is hardly any difference observed between equatorial and off equatorial latitudes. The results obtained for std(dNe) is consistent with that of Liu et. al. 2007, but std(dNe)/Ne shows discrepancies with the results of Liu et al 2007. Our study was designed to focus specifically on the geomagnetic and solar activity dependence of the seasonal and longitudinal distribution of ionospheric irregularities particularly for Figures 11 and 13 this time. However, we will consider including the results presented in Figure 1. Thank you for the suggestion.

(6) P13 Line 12, Linking word is missing.

**Response:**The formation of small-scale irregularities appears to be more likely in ionospheric regions with higher background electron density and steep electron density gradients (Keskinen et al., 2003; Muella et al., 2008; Muella et al., 2010). Therefore, the amplitudes of ionospheric irregularities closely depend on background electron density (Wan et al., 2018) and steep Ne gradient globally, as expected. This will be corrected in the revised manuscript.

[Figure]

Figure 1: The distribution characteristics of (a) std(dN e ) and (b) std(dN e )/N e with respect to 10.7 cm solar radio flux in solar flux units and kp for the period from October 2014 to October 2018.

[Figure]

Figure 2: A summary of the latitudinal ranges

(7) P17, Line 11-12: It is not an accurate description.

   **Response:**This will be rephrased to, "The $\Delta N_e/N_e$ shows a peak at the quasi-dipole equator which gradually

decreases towards higher latitudes" in the revised manuscript.

(8) In many, authors inappropriately describe the $\Delta N_e$ but not $\Delta N_e/N_e$ as the 'irregularities' (e.g. the captions of Figure 13, 14).

**Response:**The captions of Fig.14 will be rephrased to ,"Geomagnetic effect on seasonal and longitudinal distribution of $\Delta N_e$ and $\Delta N_e/N_e$ for the period from October 2014 to October 2018: A case for Swarm A." and this will also be corrected in the revised manuscript.

**Final Remark:**We thank the reviewer for the multiple comments.

---

## Author Response (AR1)

The authors of this manuscript thank the reviewers for the suggestions and comments. In enhancing the quality of the paper, all the remarks we received on this research were taken into consideration and we present our response to each of them individually below. A marked-up manuscript version has also been embedded at the end of this document.

**Response to Reviewer 1 comments**

**General Comments:**

– The results shown confirm earlier published findings, and some other analyses do not seam fully consistent (see specific comments). The submitted manuscript provides an excellent report to indicate that identified fluctuations in electron density time series obtained by the TII faceplate technique onboard the Swarm satellites with 16Hz sampling rate are most probably related to equatorial post-sunset plasma irregularities. But it is unlikely that the report enhances present scientific knowledge and evidence.

My major concern is, however, that this analysis does not show any findings based on the 16Hz samplings that could not have been achieved with the 2 Hz samplings onboard Swarm. The authors are encouraged to exploit the high value of the 16 Hz sampling.

– **Response:**

  – This study aimed at checking the capability of the Swarm 16-Hz faceplate electron density measurements for equatorial ionospheric irregularity observations at sub-kilometer scale length. The capacity of the 2 Hz electron density measurements made by the Langmuir Probes on board Swarm are not disregarded in this study. However, as stated in P.2,L.34 and presented in Figure 1 (P.5), high-resolution data enables smaller scale structures to be identified in electron density (Nishioka et al., 2011,https://doi.org/10.1029/2011ja016446). As a new aspect which is possibly relevant for especially the small-scale irregularities we have also analyzed the along-track gradient of $N_e$ (as a proxy of the pressure gradient).

  – We agree with the reviewer about exploiting further the 16 Hz Swarm electron density measurements. Previously, Alfonsi et al., 2007, http://dx.doi.org/10.1016/j.asr.2017.05.021 recommended high resolution electron density measurements as input to the WAM model. However, as stated in P.2, L.8-10, the dependence of ionospheric irregularities on the geophysical parameters, remains a problem in modeling their variation for predictive purposes (Yizengaw and Groves, 2018,https://doi.org/10.1029/2018sw001980). The results presented in this study, characterize sub-kilometer ionospheric irregularities basing on various geophysical parameters as a first step towards developing a model from Swarm 16-Hz in situ measurements of electron density. Currently, the team involved in this paper are carrying out further research involving modeling of amplitude scintillation from Swarm 16 Hz measurements using the WAM model. We thank the reviewer for the suggestion.

**Selected specific comments:**

1. **Comment:**The concept of deriving electron density from the TII faceplate technique is an extended product in the concept of the Swarm mission. A bit a more detailed description of the retrieval and the data shall be added, next to refering to "Buchert, S.: Extended EFI LP data FP release notes, ESA Technical Note, 2016." (I did not find the document in the web, and do not know if/where it is accessible.)

   **Response:** The ESA Technical report is accessible via ftp at: `ftp://swarm-diss.eo.esa.int/Advanced/Plasma_Data/16_Hz_Faceplate_plasma_density`. More detailed information about obtaining electron density from the TII can also be obtained from Knudsen et al, 2017, doi:10.1002/2016ja022571. The reference Buchert, S. 2016 has been update in the revised manuscript with the ftp link added (see Pg.22 L.21-22). In addition, the following statement has been included in the revised manuscript.

2. **Comment:**P2, L1-14: The authors have access to the full Swarm mission data to perform orbital analyses, such as the spacing of Swarm A/C or the local time processing. Alternatively, they may refer to classical papers, such as provided in doi:10.5047/eps.2013.07.001 if needed. You might reconsider, if Kil et al. or Xiong et al. are suitable references in this broad context.

   – **Response:**

      – We agree with the reviewer that it would be interesting to perform orbital analysis for the available Swarm 16 Hz data set during the period of study. Previously, Zakarenkova et al. (2016),DOI 10.1186/s40623-016-0490-5, presented orbit analysis for Swarm A and B and they stressed the local time difference between Swarm A and B during the period from August 2014 to January 2016. We present a snap shot of their result in Figure **??**. Also,

[Figure]

**Fig. 1** Temporal difference between SWA/SWB satellites from August 2014 till January 2016: **a** LT coverage during the considered seasons and LT satellite separation, **b** progressive time shift in hours between satellites. *Minutes* are indicated in decimal format. *Months* are marked by the *first letter* of their names. *Gray shadows* indicate a post-sunset interval of 18–06 LT, when equatorial plasma irregularities often occur

**Figure 1.** From Zakarenkova et al. (2016),DOI 10.1186/s40623-016-0490-5

Xiong et al (2016),DOI 10.1186/s40623-016-0502-5 presented the evolution of the longitudinal separation and time lag between the Swarm spacecrafts during the early mission phase (December 9, 2013-January 27, 2014) and the final constellation phase (April 17, 2014- September 27, 2015). We also present a snap shot of their results in Figure 9:

[Figure]

**Fig. 1** Evolution of the **a** longitudinal separation and **b** time lag between Swarm spacecrafts when they are passing the geographic equator during the early mission phase: December 09, 2013–January 22, 2014. The three spacecrafts were divided into three pairs and presented with *different colors*. During this period, the local times covered by the three spacecrafts were almost the same, which decreased from 0100 to 2100 LT for the descending node. Therefore, we only present the local time for Swarm A, as indicated by the *gray dashed line* in frame (**b**). Frames **c** and **d** are the same as **a** and **b**, but for Swarm A and C during the final constellation phase: April 17, 2014–September 27, 2015. The local time for the descending node of Swarm A has also been presented by *gray dashed line* in frame (**c**)

**Figure 2.** From Xiong et al (2016),DOI 10.1186/s40623-016-0502-5

However, as suggested by the reviewer, the five years of Swarm mission data used in the present study is sufficient for a long term orbital analysis to provide an up to date orbital status. Using five years of Swarm mission data, Claudia Stolle and the Swarm Science Team (2018), (https://cedarweb.vsp.ucar.edu/wiki/images/d/db/2018CEDAR_Monday _Stolle2.pdf) also presented orbital analysis at the CEDAR 25 June 2018 Santa Fe. A screen shot of one of the slides is shown in Figure 8. Some of the Swarm satellite orbit specifications presented by Stolle and the Swarm Science team by

[Figure]

**Figure 3.** From Stolle and the Swarm Science Team, https://cedarweb.vsp.ucar.edu/wiki/images/d/db/2018CEDAR_Monday _Stolle2.pdf

2018 are shown in a screen shot here in Figure 10.

[Figure]

**Figure 4.** From Stolle and the Swarm Science Team, https://cedarweb.vsp.ucar.edu/wiki/images/d/db/2018CEDAR_Monday _Stolle2.pdf

Basing on the reviewer's suggestion, Figure 7 shows results of orbital analysis we carried out for the period of study (October 2014 to October 2018). Note that Figure 7 presents orbital analysis results only when the 16 Hz Ne data was available during the period of study. Figure 7 presents the evolution of the longitudinal separation between Swarm A and C, Local time of Swarm A, B, and C and time lag between Swarm spacecrafts when they passed the geographic equator during the period from October 2014 to October 2018. During the period of study, the longitudinal separation between Swarm A and C varied frequently between 1.4° and 1.46°. In general, the longitudinal separation between Swarm A and C was about 1.4° corresponding to about 160 km distance at the equator, similar to what was stated by Xiong et. al. (2016), DOI 10.1186/s40623-016-0502-5 and also presented by Stolle and the Swarm team. Figure 7(b) presents the temporal difference between Swarm A/C and B satellites from October 2014 till October 2018. From Figure 7(b) , Swarm A and C cover nearly the same local time and the time lag between Swarm A and C varied frequently between 1-10 seconds (see Figure 7(c)). The local time separation

[Figure]

**Figure 5.** From Xiong et al (2016),DOI 10.1186/s40623-016-0502-5

between Swarm B and the two lower pairs is seen to increase and by October 2018, it reached up to 8 hours of time lag.

Therefore, basing on the orbit analysis results suggested by the reviewer, the sentences in Pg. 3 L. 14-20 have been updated to:

Using Swarm mission data collected from October 2014 to October 2018, orbit analysis was carried out to check on the status of the Swarm orbits. From the orbit analysis, by the end of October 2018, the longitudinal separation between Swarm A and C was about 1.4° corresponding to about 160 km distance at the equator, covering nearly the same local time sector with a time lag of about 1-10 seconds. The time lag between Swarm B and the lower pairs reached up to 8 hours. Swarm A and C were orbiting at an altitude of about 448 km altitude (orbital inclination angle of 87.35°) above sea level over the low latitude region, and Swarm B was orbiting at an altitude of about 512 km (orbital inclination angle of 87.75°).

The statement, " 'Swarm A and C need about 133 days to complete all 24 hours of local time and Swarm B needs about 141 days (Xiong et al., 2016b).' has been eliminated from the revised manuscript.

3. **Comment:**P2, L25ff: The proposed detection method identifies amplitudes of deviations from a 2s-average and attribute a level of higher than $0.25 \times 10^{10} \mathrm{m}^{-3}$ of its RMS as being an irregularity. By this method, identifying an RMS over a time window of 2 seconds (15 km), information from the 16 Hz seconds are smoothed out.

– **Response:** The window to calculate $dN_e$ (for the case of small-scale ionospheric irregularities) should be short, but has to be long enough to avoid spurious detection of ionospheric irregularities. We also tried 1 s, 16 points, instead of 2s, 32 points, and the outcome was similar and reasonable. In addition, during the study period, the solar activity was generally low (descending phase of solar cycle 24), and ionospheric plasma perturbations were, in general, weak. Therefore, we chose a smaller threshold to capture a reasonable number of events.

4. **Comment:** Figure1/P5, L3: 'multi-peak' variations appropriate to the 16 Hz samples in comparison to the 2Hz samples cannot be identified in either the zoomed figures in Figure 1. Many peaks are equally visible in the 2 Hz data.

– **Response:** We present here in Figure 6 inset plots which are zoomed in further. It can now be clearly seen that multiple

[Figure]

[Figure]

**Figure 6.** Comparison of 2 Hz LP data and 16 Hz faceplate data for Swarm A and C on 2014-10-06 and 2015-07-03, respectively.

electron density peaks are observed on the 16 Hz data than in the 2 Hz data. The 2 Hz data appears to be smooth and cannot show detailed small scale structures. Figure 1 in Pg. 5 in the revised manuscript has been replaced with the new zoomed in Figure.

5. **Comment:** Page 10,L9: "Large values of both $\Delta N_e$ and $\Delta N_e/N_e$ often occur in locations of large depletions in Ne." It is not clear, where else they could be expected?

– **Response:** In the low latitudes, the values of both parameters are expected to be high in locations of plasma density depletions at the EIA crests, but also at the bottom of large scale bubbles/depletions. However, the absolute perturbation is high at the crests and at the edge of large scale bubbles/depletions. So high values of both the absolute and relative perturbations do not occur at the same place. We have shown this for the crests/equator in Figures 6, 7, and 8. To make it clear, we have rephrased the sentence to:

From Fig.2 panels (b) and (c), large values of both $\mathrm{std}(dN_e)$ and $\mathrm{std}(dN_e)/N_e$ often occur in locations of large depletions in $N_e$ at the EIA crests close to the quasi-dipole equator, but also at the bottom of large scale bubbles.
(see Pg. 5 L 14-15, Pg. 6 L1.)

6. **Comment:** Page 5, L15-23: The effect of relative variation compared to absolute variation was extensively discussed, e.g., for polar patches in DOI: 10.1002/2017JA024811 .

– **Response:** Chartier et al. (2018) DOI: 10.1002/2017JA024811 used $N_e$ and Total Electron Content (TEC) measurements made by the LPs and GPS, respectively on board Swarm to test the relative and absolute perturbations in detection of polar cap patches. In the present manuscript, we extended the comparison of relative and absolute variation in the low

latitude region using the polar orbiting Swarm satellites, where the distribution of ionospheric irregularities seems to be quite different from that at the polar region. In addition, Chartier et al. (2018) used Swarm LP $N_e$ measurements at a frequency of 2 Hz, while in the current study we used the high resolution faceplate measurements. To acknowledge the work presented by Chartier et al. 2018, DOI: 10.1002/2017JA024811, the following sentences have been added in the revised manuscript:

Chartier et al. (2018) used Ne and Total Electron Content (TEC) measurements made by the LPs and GPS, respectively on board Swarm to test the relative and absolute perturbations in detection of polar cap patches. In terms of seasonal distribution, they observed discrepancies between the two methods with relative disturbances showing more patches in winter than in summer. However, the study presented by Chartier et al. (2018) was limited to the high latitudes.
(see Pg. 4 L. 1-5)

7. **Comment:** P7, L3: The authors describe that the irregularities occur between 18-06 LT, however, no other LT is shown in Figure 3.

– **Response:** The local time sector was fixed between 1800-0600 LT since over the low latitude region, ionospheric irregularities are a night time phenomena and this has been stated in the manuscript on Page 1, L. 20-21 and Page 6, L. 16-18. The reviewer is referred to Kil and Heelis, 1998; Burke et al., 2004; Su et al., 2006; Stolle et al., 2006; Dao et al., 2011; Huang et al., 2014; Xiong et al., 2016b; Wan et al., 2018 (also cited in Page 6, L. 16-18). Most of these earlier studies also limited their analysis over the low latitude region to the local time sector 1800-0600 LT. To make this clear, the statement has been rephrased to:

As mentioned earlier, ionospheric irregularities in the low latitudes are a nighttime phenomena and therefore, the analysis was restricted to the time period from 1800 LT-0600 LT. From Fig. 3, it is seen that irregularities occur from 1800 LT to 0600 LT as expected, irrespective of the method used.
(see page 6, L. 25-26, page 7, L. 1)

8. **Comment:** Figures 6,7,12 base on color scales that lie below the detection threshold by at least to 50%

– **Response:** For the climatology maps presented in Figures 6,7,12,14, all Swarm passes were considered irrespective of whether there were irregularities or not. This was done because of the limited data. However, the plots were regenerated using only Swarm passes which encountered irregularities. The results are presented in the revised manuscript. Although, less data was used after applying the defined filter, the results are reasonable and the color scales have also been updated in this regard (see Figure 6,7,12,14).

9. **Comment:** Figure 9 shows along track gradients of electron density, that are by nature directly related to $\Delta N_e$ for the detections. It is maybe not helpful to discuss their coincidence.

– **Response:** As the reviewer has mentioned, the along track gradients are indeed directly related to std($dN_e$). However, technically, the $\nabla N_e$ is the slope of a fitted straight line to the data, and std($dN_e$) is from the residuals. Both are independent, i.e. the method does not automatically imply a correlation between both. An observation of interest was that there was a slight latitudinal difference between std($dN_e$)/$N_e$ and $\nabla N_e$ and therefore, it was important to include $\nabla N_e$ and compare it to std($dN_e$)/$N_e$ and std($dN_e$). Also, from the best of our knowledge, seasonal dependence of latitudinal-longitudinal distribution of $\nabla N_e$ has not yet been presented for the case of the Swarm satellites.

10. **Comment:** Figure 12 divides below and above F10.7=140sfu. It is questionable if a significant amount of data is available for conclusions above F10.7=140sfu. See figures 10, 11 of the submission and https://services.swpc.noaa.gov/images/solar-cycle-10-cm-radio- flux.gif. Also, P 14 L 20ff: The authors mention similar comparisons by Huang et al., Su et al., Stolle et al. To the knowledge of the referee, these papers discussed the occurrence rate, not the amplitude of irregularities. Amplitudes are discussed by Wan et al., 2018. However, conclusion (iv) of the submitted paper mentions occurrence rates which is not compatible with Figure 12.

- **Response:**

  - For the case of Fig. 12, we agree with the reviewer that the data may not have been enough after categorizing with respect to $F10.7$ especially for $140 \leq F10.7 < 180$. This limitation was also stated in the manuscript in Page 16,L 8-9 and Page 19,L 10-12. However, irrespective of the limitation in data availability, std($dN_e$) shows dependence on moderate solar activity similar to what was presented in earlier studies (e.g, Huang et al. (2001), Su et al. (2006), Stolle et al. (2006)) i.e, high std($dN_e$) values are often observed when $140 \leq F10.7 < 180$.

  - As pointed out by the referee, it is true that Huang et al. (2001), Su et al. (2006), Stolle et al. (2006) did not present amplitudes of electron density perturbation, and amplitudes are discussed by Wan et al. 2018. However, Wan et al. 2018 did not present the dependence of amplitudes of ionospheric irregularities to different solar activity levels categorized in terms of $F10.7$, while the cited papers (Huang et al. (2001), Su et al. (2006), Stolle et al. (2006)) addressed the solar activity dependence of occurrence of ionospheric irregularities. A key difference between occurrence of ionospheric irregularities and their amplitudes stated by Wan et al. 2018 is that they show a totally different longitudinal pattern. The amplitudes presented in Fig. 12 seem to show similar dependence on different levels of $F10.7$ as the occurrence rates presented by Huang et al. (2001), Su et al. (2006), Stolle et al. (2006) with high std($dN_e$) values (or high occurrence rates) often observed when $140 \leq F10.7 < 180$. Therefore, the dependence of amplitudes, std($dN_e$) and occurrence rate of ionospheric ionospheric show similar dependence on solar activity level.

    For clarification, the following statement has been added in the revised manuscript:

    It is necessary to note that Huang et al. (2001), Su et al. (2006), Stolle et al. (2006) addressed the solar activity dependence of the occurrence rate of ionospheric irregularities. Wan et. al. 2018 presented differences between the occurrence rate of ionospheric irregularities and their amplitudes in terms of the latitudinal and longitudinal distribution. However, in general, the seasonal and longitudinal distribution of std($dN_e$) presented in Fig. 12 shows a similar dependence on different levels of F10.7 as the occurrence rates.
    (see Pg. 17 L. 1-5)

  - Conclusion (iv) has been rephrased considering a suggestion from Reviewer 2 Comment (5) to,

    The std($dN_e$) showed a weak positive correlation with F10.7 and the correlation was even lower with std($dNe$))/$N_e$. Furthermore, std($dNe$)) in the EIA crest regions grew approximately linearly from the low to moderate solar activity, with higher correlation than that in the EIA trough region. The F10.7 dependence of the seasonal and longitudinal distribution of the ionospheric irregularities showed slightly different trends between std($dNe$)) and std($dNe$))/$N_e$. The discrepancy between the two methods may be because of the limited data. In general, the distribution of ionospheric irregularities was still lower during the geomagnetically disturbed period than in quiet times. Close inspection of showed that the correlation between std($dNe$)) and Kp was lowest at the EIA belts compared to that at the equator. The solar and magnetic activity dependence of std($dNe$))/$N_e$ hardly showed any difference in correlation between equatorial and off-equatorial latitudes.
    (see Pg.20 L.26-33)

11. **Comment:** Palmroth et al., 2000 and Stolle et al., 2006 did first analyses on the relation between the occurrence rate of irregularities and the Kp index, that are not discussed in the submitted manuscript.

  - **Response:** In the updated manuscript, the literature recognized by the reviewer have been integrated. (see Pg. 20 L. 11 and Pg. 19 L. 1-6)

12. **Comment:** The authors might reconsider the added value of P11 L6ff to the concerned study.

  - **Response:** Concerning the difference between Swarm A/C and Swarm B, orbital analysis such as spacing between Swarm A and C, altitude difference between Swarm A/C and B, time lag between Swarm A/C and B etc was carried out in response to Comment 2.

**Selected technical comment:** dNe, $\Delta N_e$, and (nabla)Ne are used to express the same parameter.

– **Response:**

   – In the revised manuscript, we have replaced $\Delta N_e$ with std($dN_e$) and std($dNe$))/$N_e$ to represent the absolute electron density perturbation. Also, $\Delta N_e/N_e$ has been replace with std($dNe$))/$N_e$ to represent the relative electron density perturbation. The std($dN_e$) is obtained by determining the standard deviation of the residual $dN_e = N_e - \overline{N_e}$.

   – $\nabla N_e$ in this manuscript represents the electron density gradient derived along the satellite tracks and it is given by: $\nabla N_e = \frac{(N_e)_f - (N_e)_i}{X_f - X_i}$, where $X$ in this formula represents the latitude, $f$ is the final position and $i$ is the initial position.

**Response to Reviewer 2 Comments**

**Comment:** First of all, I would like to make some clarification according to my understanding of the manuscript. 1. The small-scale irregularities are usually embedded in the large scale irregularities (seen Figure 2 and from my experience), and they are actually the regional density fluctuation inside the plasma bubbles (PBs). 2. According to the authors' definition on the $\Delta N_e$, that is, the standard deviation of the residuals between the original data with the mean fitting, in 32 data points (2 seconds, 0.6 degrees in latitudes, 14 Km in length). The $\Delta N_e$ quantified the density fluctuation in a spatial scale of 0.6° GLAT (normally inside a major PB), which is different from the conception of the depletion amplitudes, in my opinion.

**Response:**(1.) We agree with the reviewer and we are aware of this fact.(2.) We also agree that the concept of a depletion amplitude is less applicable for the small-scale amplitudes analyzed in this work, and the submitted text is not clear about this. The revision has been improved in this respect (confirm from Response to Major comments 1 and 2). As a new aspect which is possibly relevant for especially the small-scale irregularities we also analyzed the along-track gradient of Ne (as a proxy of the pressure gradient).

**Major Comments**

(1) **Comment:**The author should emphasize in the article that these small-scale irregularities is not independent and has not been distinguished, from those of large-scale (PBs).

**Response:** The following sentence have been included in the revised manuscript,
Here, we mostly dealt with small scale ionospheric irregularities which are relevant for L-band scintillations. It is essential to note that these small-scale irregularities are not autonomous from those of large-scale ionospheric irregularities and they were not differentiated. The only satellite sampling the bottom-side, at altitudes below 300 km, has so far been the Atmosphere Explorer-E (AE-E) (Kil and Heelis, 1998). With the AE-E satellite, irregularities with relatively small amplitudes were seen without clearly developed EPBs. At altitudes above 350 km addressed by nearly all other studies smaller scale irregularities are usually embedded in EPBs. Automatic detection algorithms used in previous works do not discriminate between depletions (EPBs) and irregular multi-peak variations, with the exception of Wan et al. (2018), whose algorithm determined a depletion amplitude.
(see Pg, 4, L. 25-32).

(2) **Comment:**Authors compared the occurrence of scintillations (quantified by ROTI) in 9 stations with the small-scale irregularities which exhibit correlations. However, as noted in previous works (e.g. Xiong et al., 2016; Wan et al., 2018), the radio signal disruption is more likely to occur when there exists PBs with large depletion amplitude. In the same reason as described in my first clarified issue, I think authors should be careful to relate the two things up.

**Response:**

We are aware of the fact that radio signal signal disruption is more likely to occur when there exists PBs with large depletions. We also sated in Page 2, L 11-12 that an interesting feature of these plasma irregularities is their scale sizes. They typically cover a variety of scale sizes, from a few centimeters to thousands of kilometers (Zargham and Seyler, 1989 , https://doi.org/10.1029/ja094ia07p09009 ; Hysell and Seyler, 1998,https://doi.org/10.1029/98ja02616)". The bubbles themselves have horizontal sizes of order 100 km. The 9 IGS stations used in this study transmit radio signals at two L-band frequencies i.e, L1 (1575.42 MHz) and L2 (1227.60 MHz). The small-scale electron density variations commonly termed as small-scale structures are known to affect more L-band transmissions (Luhr et. al. 2014, doi: 10.3389/fphy.2014.00015, Bhattacharyya et. al. 2003, doi:10.1029/2002RS002711, Rao et. al. 2005, doi:10.1007 / s00585-997-0729-3, Spogli et. al. 2016, doi:10.1002/2016JA023222). The Swarm high resolution 16 Hz density estimates correspond to a spatial resolution of about 500 m, which is within the range of applicable Fresnel scales, and so theoretically relevant as a cause of L-band scintillations. Therefore, in the current manuscript, we compared small scale irregularity structures quantified from the Swarm 16 Hz electron density measurements with ROTI derived from VTEC measurements at L-band frequency.

Furthermore, the small scale irregularities in the equatorial topside ionosphere are mostly imbedded in EPBs, but generally they can exist independently of them. According to Kil and Heelis (1998), https://doi.org/10.1029/97JA02698 at the bottom-side, below 300 km, small scale irregularities were frequently seen after sunset, but they rarely developed EPBs. At high latitudes small scale irregularities are often associated with polar patches which are structures of increased electron density. We acknowledge that Wan et al. (2018), https://doi.org/10.1002/2017JA025072 developed an algorithm to determine a depletion amplitude, which should be sensitive to only EPBs. However, scintillations of radio signals would be caused by irregularities broadly near the Fresnel scale, about 500 m for the L band, and not directly by the larger scale EPB depletions. This is at least according to the well-know phase screen model. We suspect that the result of Wan et al. (2018), https://doi.org/10.1002/2017JA025072 could be explained because deep EPBs involve steep density gradients which would create the small scale irregularities. Indeed we find a correlation between std($dN_e$) and the along track $\nabla N_e$ and this is presented here in Fig 7. In the equatorial F region the large amplitude EPBs are the major source of small scale irregularities

[Figure]

**Figure 7.** Correlation between std($dN_e$) and $\nabla N_e$.

but not necessarily the only one. Therefore we have based our statistical analysis directly on 16 Hz (500 m scale) variations, irrespective of their creation mechanism.

(3) **Comment:** P13 Line10 – 12: Obviously, the s/l variation of $\nabla N_e$ is similar to std($dN_e$) but not std($dN_e/N_e$), authors should describe the plots objectively.

**Response:** The following description has been included in the manuscript in Pg. 14 L.10-15 and Pg. 15 L.1-4
The seasonal and longitudinal distribution of $\nabla N_e$ generally shows the same pattern as that of std($dNe$) and std($dN_e$)/$N_e$ with high values observed during the equinoxes and December solstice and moderate values in the African sector in June solstice. However, close inspection of Figure 9 shows that the $\nabla N_e$ has the same latitudinal distribution as std($dN_e$) i.e., it is symmetrical about the magnetic equator with high values at the EIA belts. On the other hand, the latitudinal distribution of $\nabla N_e$ is different from that of std($dN_e/N_e$). Earlier studies have also shown that irregularity events at latitudes of the EIA might be associated with the regions of strong density gradient (e.g, Basu et al., 2001; Keskinen et al., 2003). Therefore, the formation of small-scale irregularities appears to be more likely in ionospheric regions with higher background electron density and steep electron density gradients (Keskinen et al. 2003; Muella et al. 2008; Muella et al. 2010).

**Minor comments**:

(1) **Comment:**Figure 1: I would recommend the author to plot all the profiles using the absolute scale.

**Response:**Using the logarithmic scale makes it clearer, that the 2 Hz and 16 Hz data differ by only a nearly constant factor (which is of little relevance when analysing variations of $N_e$). We present here a comparison for Swarm A on 2014-10-06. As a response to a concern raised by reviewer 1 about multi-peak variations not observed in zoomed-in

section, we have zoomed in further in Figure 1 and maintained the logarithmic scale.

(2) **Comment:**P8, L19 - 20: in pre-midnight sector.

**Response:** In P8, L20-P9, L2, we were describing the enhanced post-midnight irregularities seen in Fig. 2(b) for June Solstice. To make it clear, the sentence has been rephrased to,

The percentage occurrence in June Solstice is generally small, comparable to that observed in Fig.2, with a broad plateau extending post-midnight. However, the enhanced post-midnight irregularities seen in Fig. 2(b) for std(dNe/Ne) in June Solstice are not observed in the LT trend of ROTI. Therefore, the LT dependence of percentage occurrence of ionospheric irregularities quantified using std(dN e) closely follows the same trend as that of ROTI for all seasons.
(see Pg. 9 L.21-22, Pg. 10 L.2)

(3) **Comment:**P8, L20 – P9, L2: It's not appropriate to explain that in a decayed sense, since the occurrence of irregularities at post-midnight is enhanced as authors found and claimed, the mechanism should be related to the lower depletion amplitudes.

**Response:** We do not claim that the enhanced irregularities post-midnight is related to lower depletions. The mechanisms that generate these post-midnight irregularities are still unknown and widely debated. Two mechanisms to explain post-midnight irregularity have been suggested. One mechanism is the seeding of the RTI by atmospheric gravitational waves from below into the ionosphere, while the other mechanism is the elevation of the F-layer by the thermosphere's meridional neutral winds, which may be connected with the thermosphere's highest midnight temperature. The following sentences have been included in the revised manuscript as suggested explanations for the post-midnight enhancement observed:

The mechanisms that generate these post-midnight irregularities are still unknown and widely debated. Two mechanisms to explain post-midnight irregularity have been suggested. One mechanism is the seeding of the RTI by atmospheric gravitational waves from below into the ionosphere, while the other mechanism is the elevation of the F-layer by the

thermosphere's meridional neutral winds, which may be connected with the thermosphere's highest midnight temperature (Otsuka, 2018, and references therein). (see Pg.8 L.12-13, Pg.9 L.1-2)

The statement, 'Equatorial ionospheric irregularities develop just after sunset under favorable conditions and then decay as time progresses (Kil et al., 2009)." has been removed from the manuscript.

(4) **Comment:**Figure 5b: It seems that the scintillation data is less available during June Solstice compared to other seasons for some stations. E.g. RIOP is totally missing. The author should mention those in the article.

**Response:**The following sentence has been included in the revised manuscript in Pg.9 L.16-17:

It is important to note that RIOP did not have data in June solstice and September equinox as seen in panels (b) and (c) of Fig. 5.

(5) **Comment:** Figure 11& Figure 13: I suggest the authors do the correlation analysis regards to different latitudinal bins (i.e. equator and off-equator).

**Response:**We agree with the reviewer that the geomagnetic and solar activity dependence of occurrence of ionospheric irregularities at different latitudinal regions (i.e, equator and off-equator) would be interesting to know.

- Previously, Liu et. al. 2007, JGR,112, A11311, doi:10.1029/2007JA012616 investigated the solar activity dependence of the electron density at equatorial and low latitudes. We present here a snapshot of one of the results extracted from Liu et. al. 2007 concerning latitudinal distribution on solar activity dependence of electron density. From the results presented by Liu et. al. 2007, the electron density in the crest regions of the EIA grows roughly

[Figure]

**Figure 8.** Same as Figure 7 but for postsunset (1800−2300 MLT) sector for the (left) northern EIA crest, (middle) dip equator, and (right) southern EIA crest.

linearly from solar minimum to solar maximum, with higher growth rate than in the EIA trough region. However, Liu et. al. 2007 did not quantify ionospheric irregularities in their analysis. Therefore, we attempted the suggestion made by the reviewer.

- The Swarm satellite passes were divided into three latitudinal ranges i.e, Equator ($\pm5°$ quasi-dipole latitude), Southern EIA region (-30°− -5° quasi-dipole latitude), and Northern EIA region (+5°−+30° quasi-dipole latitude). A summary of the latitudinal ranges is presented in a map in Figure 8.

[Figure]

**Figure 8.** A summary of the latitudinal ranges

Scatter plots of (a) std($dN_e$) and (b) std($dN_e$)/$N_e$ as functions of F10.7 are shown in Fig. 9 for Swarm A, B, and C, independently. The correlation is generally low irrespective of the latitudinal region and this may be attributed to the small data sample. However, it can be seen that the correlation between F10.7 and std($dN_e$) is higher at the EIA regions than at the equator. This shows also the symmetrical distribution of std($dN_e$) with high values obtained at the EIA belts than at the equator as seen in Fig. 6 in the manuscript. On the other hand std($dN_e$)/$N_e$ hardly shows any significant difference in correlation with F10.7 at the equator and at the EIA belts. For the case of std($dN_e$) dependence on Kp as presented in Fig. 10, it is a generally weak correlation and it is more negative at the equator than off equatorial latitudes. For std($dN_e$)/$N_e$, the correlation is very weak with both kp and F10.7. There is hardly any difference observed between equatorial and off equatorial latitudes. The results obtained for std($dN_e$) is consistent with that of Liu et. al. 2007, but std($dN_e$)/$N_e$ shows discrepancies with the results of Liu et al 2007.

- The reviewers suggestion has been considered in the revised manuscript and Fig 11 and 13 in the manuscript have been replace with Fig 2 and Fig 3. Also the following descriptions have been incorporated in the manuscript:
  To check on the solar activity dependence of std($dN_e$) and std($dN_e$)/$N_e$ at the equator and the EIA belts, the Swarm satellite passes were divided into three latitudinal ranges i.e, Equator ($\pm5°$ quasi-dipole latitude), Southern EIA region (-30°$-$ -5° quasi-dipole latitude), and Northern EIA region (+5°$-$+30° quasi-dipole latitude) (see legend of Fig. 11). Each panel of Fig. 11 contains linear fits and the correlation coefficients $R$. In general, both std($dN_e$) and std($dN_e$)/$N_e$ show weak positive correlation with F10.7 irrespective of the latitudinal range and this may be attributed to the small data-set used. However, it can be seen that the correlation between F10.7 and std($dN_e$) is higher at the EIA regions than at the equator. This also shows the symmetrical distribution of std($dN_e$) with high values obtained at the EIA belts than at the equator. There is hardly any difference observed between equatorial and off equatorial latitudes for the case of std($dN_e$)/$N_e$ . The results obtained for std($dN_e$) is consistent with that of Liu et al (2007) who presented the solar activity dependence of the electron density in the equatorial anomaly regions.(see Pg 15 L. 18-23, Pg 16 L. 1-4)
  To generate Fig. 13, the Swarm passes were also split into equatorial and EIA latitudes, similar to Fig. 11. (see Pg 17 L. 10-11)
  Close inspection of Fig. 13 shows that the correlation between std($dN_e$) and Kp was lowest at the EIA belts compared to that at the equator. There is hardly any difference observed between equatorial and off equatorial latitudes for the case of std($dN_e$)/$N_e$ . Dao et al. (2011) adopted the relative perturbation to quantify irregularities. Their reason for using the relative perturbation was that the absolute perturbation is correlated with the ambient ion density, which varies due to several factors such as varying altitude. The results shown in Fig. 11 and 13 also show that std($dN_e$) is more sensitive to solar and magnetic variations compared to std($dN_e$)/$N_e$ . The differences in the correlation between F10.7 or Kp and std($dN_e$) or std($dN_e$)/$N_e$ at equatorial and off-

**(a) std($dN_e$)**

[Figure]

**Figure 9.** The distribution characteristics of (a) std(dN e ) and (b) std(dNe)/N e with respect to 10.7 cm solar radio flux in solar flux units for the period from October 2014 to October 2018.

equatorial latitudes can be explained by the differences in background electron density and electron density gradients at the crests and trough.(see Pg 17 L. 13-14 and Pg 18, L. 5-6)

Conclusion (iv) has also been re-written to:

The std($dN_e$) showed a weak positive correlation with $F10.7$ and the correlation was even lower with std($dN_e$)/$N_e$. Furthermore, std($dN_e$) in the EIA crest regions grew approximately linearly from the low to moderate solar activity, with higher correlation than that in the EIA trough region. The F10.7 dependence of the seasonal and longitudinal distribution of the ionospheric irregularities showed slightly different trends between std($dN_e$) and std($dN_e$)/$N_e$. The discrepancy between the two methods may be because of the limited data. In general, the distribution of ionospheric irregularities was still lower during the geomagnetically disturbed period than in quiet times. The correlation between std($dN_e$) and Kp was lowest at the EIA belts compared to that at the equator. The solar and magnetic activity dependence of std($dN_e$)/$N_e$ hardly showed any difference in correlation between equatorial and off-equatorial latitudes.(see Pg. 20 L.26-33)

L13-16 of the abstract has also been rephrased to:

The reliance of std($dN_e$)/$N_e$ on solar and magnetic activity showed little distinction in the correlation between equatorial and off-equatorial latitudes, whereas std($dN_e$) showed significant differences. With regard to

**(a) std($dN_e$)**

**(a) std($dN_e$)/$N_e$**

**Figure 10.** The distribution characteristics of (a) std(dN e ) and (b) std(dN e )/N e with respect to Kp for the period from October 2014 to October 2018.

seasonal and longitudinal distribution, high std($dN_e$) and std($dN_e$)/$N_e$ values were often found during quiet magnetic periods compared to magnetically disturbed periods. The std($dN_e$) increased approximately linearly from low to moderate solar activity.

(6) **Comment:** P13 Line 12, Linking word is missing.

**Response:** The statements have been rephrased in the revised manuscript to,
The formation of small-scale irregularities appears to be more likely in ionospheric regions with higher background electron density and steep electron density gradients (Keskinen et al., 2003; Muella et al., 2008; Muella et al., 2010). Therefore, the amplitudes of ionospheric irregularities closely depend on background electron density (Wan et al., 2018) and steep Ne gradient globally as expected.
(see lines Pg. 15 L.1-4)

(7) **Comment:** P17, Line 11-12: It is not an accurate description.

**Response:** This has been rephrased to,
The std($dN_e$)/$N_e$ showed a peak at the quasi-dipole equator which gradually decreased towards higher latitudes.
in the revised manuscript. (see Pg.20 L.20-21)

(8) **Comment:**In many, authors inappropriately describe the $\Delta N_e$ but not $\Delta N_e / N_e$ as the 'irregularities' (e.g. the captions of Figure 13, 14).

**Response:**The captions of Fig.13 and Fig. 14 have been rephrased to ,
The distribution characteristics of (a) $\text{std}(dN_e)$ and (a) $\text{std}(dN_e)/N_e$ with respect to Kp index for the period from October 2014 to October 2018. The black lines in each panel represent a linear fit to the data.
Geomagnetic effect on seasonal and longitudinal distribution of $\text{std}(dN_e)$ and $\text{std}(dN_e)/N_e$ for the period from October 2014 to October 2018: A case for Swarm A.,
respectively in the revised manuscript.

**Response to Reviewer 3 Comments**

In this paper, the ionospheric irregulariteis are studied by using two methods: the absolute and relative changes of electron density. The distribution characteristics with local time, season and longitude of irregular structures are reproduced. The differences in the latitudinal distribution, and the correlation difference with F107 between the two methods are reported. The main problem for the reviewer is that the relative change depends on both the absolute change in electron density and the value of background electron density. The strong relative change in the equatorial region may be due to the weaker background electron density. The dependence on F107 is not strong and may also be due to changes in background electron density with F107. From this point of view, perhaps absolute changes may be more suitable for studying the ionospheric irregularities. So I don't know why the author should use the relative change as a way to study the ionospheric irregularities? I hope they can explain why. More interesting is the finding of the importance of the meridional difference in electron density in the irregular structure of the ionosphere. Could they explain more about the physical mechanism?

**Response:**

– We are aware of the fact that the relative change depends on both the absolute change in electron density and the value of the background electron density. We are also aware that the strong relative change in the equatorial region may be due to the weaker background electron density. We also mentioned in the manuscript in Pg. 14 L. 3-5 that, "From both local time and longitudinal perspectives, Wan et al. (2018), https://doi.org/10.1002/2017JA025072 confirmed that the depletion amplitudes of irregularities are closely linked to the background electron density intensity." From the results obtained by Wan et al. (2018), https://doi.org/10.1002/2017JA025072 it is observed that high background electron density is concentrated at the EIA belts than at the magnetic equator. We also mentioned in Pg. 14 L.1-3 that," 'Among other parameters, the growth rate of equatorial ionospheric irregularities is controlled by the electron density gradient. Ionospheric irregularities in the equatorial and low latitudes can cascade upwards and along the magnetic field lines to the EIA belts characterized by high background $N_e$ and steep gradients in density (Muella et al., 2010).'

– The question asked by the reviewer i.e, why we use the relative change as a way to study the ionospheric irregularities, is one of the motivations for the current manuscript. Multiple studies have adopted the relative change to quantify ionospheric irregularities, while others have adopted the absolute perturbation (The reviewer is referred to the literature cited in Pg.3 L.25-26). Dao et. al. (2011), doi:10.1029/2011GL047046 adopted the relative perturbation to quantify irregularities. Their reason for using the relative perturbation was that the absolute perturbation is correlated with the ambient ion density, which varies due to several factors such as varying altitude. Instead, they computed the normalized depletion of ambient density, $\Delta N/N$, in an attempt to decouple any variables associated with the varying ambient density.

  However, discrepancies between the two methods in terms of identifying ionospheric irregularities have also been identified in earlier studies (This has been discussed in Pg. 3 L. 28-33, Pg. 4 L. 1-5). In Pg.3 L. 28-33, we also explain how comparison of the two methods is important in the meridional direction. Therefore, in this manuscript we use the relative change as a way to identify ionospheric irregularities to compare with absolute change.

  The absolute changes are most relevant to assess the effects of irregularities on radio signals and scintillations, which is for applied purposes. However, for studying the physical mechanisms causing irregularities the relative change is perhaps more important. For example, if a plasma instability is suspected to be the cause of irregularities, the observed relative change can give indications whether the disturbances can be assumed linear (small relative changes up to perhaps 10 %) or non-linearity needs to be considered (larger relative changes). Therefore, we think that it makes sense to investigate the statistical occurrences of both absolute and relative changes.

– Concerning the mechanism responsible for the meridional distribution of ionospheric irregularities,

  • Ionospheric irregularities are generated after sunset in the low latitude region due to the plasma instabilities and the most important parameter for their development is the equatorial evening vertical plasma drift ($\boldsymbol{E} \times \boldsymbol{B}/B^2$) (Fejer et al,1999, https://doi.org/10.1029/1999JA900271, Abdu 2005,DOI: 10.1016/j.asr.2005.03.150)

known as the pre-reversal enhancement in vertical drift when the eastward electric field is intensified due to the action of the F region dynamo. At low latitudes, the ionosphere presents the EIA with high electron density observed between about $\pm15° - \pm20°$ magnetic latitude. The EIA have their origin in the upward $\boldsymbol{E} \times \boldsymbol{B}$ plasma drift of the equatorial F layer. The zonal electric field that exists in the equatorial ionosphere is

[Figure]

**Figure 11.** Appleton Anomaly Scheme

directed to the east during day, creating an upward vertical $\boldsymbol{E} \times \boldsymbol{B}/B^2$ drift velocity. Soon after the sunset, this eastward electric field is intensified (pre-reversal peak) by the F region dynamo and the plasma from F region is uplifted to high altitudes. Meanwhile, the plasma from low altitudes quickly decline due tp the decreasing of the intensity of incident solar radiation(Kelley, 2009). After lifting to high altitudes in the equatorial region, the plasma starts a descent movement along magnetic field lines. This movement happens due to the action of gravity (g) and pressure gradient ($\nabla p$) forces as illustrated in Figure 11. This phenomenon (the plasma elevation and the subsequent descent along magnetic field lines to low latitudes) is known as the fountain effect, giving origin to the EIA. The upward vertical plasma drift in the equator after sunset that gives origin to the pre-reversal peak, is the main factor responsible for the plasma irregularity generation (Fejer et al., 1999, https://doi.org/10.1029/1999JA900271). This means that around sunset at low latitudes, the enhanced eastward electric field in F region enlarges the equatorial fountain effect, causing the two crests of the EIA to get stronger and the trough above the dip equator to become deeper. We also quoted in Pg 12 L.10-11 that, " Ionospheric irregularities in the equatorial and low latitudes can cascade upwards and along the magnetic field lines to the EIA belts characterized by high background $N_e$ and steep gradients in density (Muella et al., 2010,https://doi.org/10.1029/2009JA014788)."

- The difference in latitudinal distribution of ionospheric irregularities as observed in the meridional direction characterized by std($dN_e$) and std($dN_e$)/$N_e$ as discussed in the manuscript can be explained by the differences in background electron density and electron density gradients at the crests and trough. As mentioned by the reviewer, high values of std($dN_e$)/$N_e$ occur at the equator because it depends on the background electron density which is low at the magnetic equator. High values of std($dN_e$) occur at the crests because of high background electron density at these locations and steep electron density gradients (as presented in Figure 9).

[revised manuscript text omitted]

---

## Referee Report (RR1)

Second review report for "Traits of sub-kilometer F-region irregularities as seen with the Swarm satellites" authored by S. Aol et al.

Authors had answered all the comments and modified the paper properly. Still, I have a minor suggestion to add the reply of my second comment in the manuscript to make it more robust. And I suggest the authors to do a clearer correlation analysis between std(dNe) or std(dNe)/Ne with Kp, more criteria like local time, seasons, solar flux could be set. If there are still no correlations, authors could simply mention the results in the manuscript.

Minor comments:
(1) P8, Line 10: should be "..occurrence at post-midnight..".

---

## Referee Report (RR2)

**Traits of sub-kilometer F-region irregularities as seen with the Swarm satellites - Manuscript review**

December 17, 2019

**General comment**

Provided manuscript focuses on the the distribution characteristics of ionospheric F-region irregularities in the low latitudes. The Authors have replied to previous comments of Reviewers and improved quality of the paper. The study uses 16 Hz electron density observations made by the faceplate on board Swarm satellites of the European Space Agency (ESA). Extensive analysis focuses on absolute (std(dNe)) and relative (std(dNe)/Ne) density perturbations, but in my opinion presented work does not fully exploits capabilities of the Swarm mission. Keeping in mind that Swarm is a mission dedicated to the Earth's magnetic field, it would be much more interesting from the scientific point of view, if the Authors could provide joint analysis of electron density and magnetic field perturbations.

It is unnecessary to perform joint analysis from the scratch, since that would highly affect the whole concept and structure of the manuscript. But Swarm provides the Level 2 data product dedicated to plasma irregularities, the ionospheric bubble index. Please visit Swarm repository to download the data:

`https://swarm-diss.eo.esa.int/#swarm%2FLevel2daily%2FLatest_baselines%2FIBI`

Taking example of discussion of irregularity structures observed by Swarm, it would be interesting to how these structures are classified by the plasma bubbles detection algorithm. For instance analysis of Bubble Flag in the product, could provide information on accompanied fluctuations in the Swarm magnetic field registrations.

---

## Author Response (AR2)

**"Traits of sub-kilometer F-region irregularities as seen with the Swarm Satellites"**

by Aol et al.

Dear Dr. Petr Pisoft,

Thank you for your letter. As authors of the manuscript angeo-2019-50, we thank the reviewers for their constructive suggestions and comments. In enhancing the quality of the paper, all the remarks we received on this research were taken into consideration and we present our response to each of them individually below. A marked-up manuscript version has also been embedded at the end of this document. We hope, our manuscript is acceptable for Annalese Geophysicae in this form.

Best regards

Sharon Aol
* * *
Below you find our point-by-point reply. For the convenience of the referees we have repeated in the response the relevant comments and then given text highlights in the revised manuscript in blue.

**Response to Reviewer 1 comments**

**Comment:** The authors have responded to all comments of all referees, and the paper has improved. There are still significant lacks, since many of the responses did not clear the issues raised. The submitted article is a very good report on the detectability of plasma irregularities in the 16 Hz data, but it is unlikely that the report enhances present scientific knowledge and evidence.

**Response:** Thank you for your valuable feedback and we are glad that our paper has improved basing on previous reviewer comments and suggestions. Below are points we wanted to address in this study:

– This study aimed at checking the capability of the Swarm 16-Hz faceplate electron density measurements for small-scale ionospheric irregularity observations. As mentioned in the response to reviewer's previous comments, the results presented in this study characterize small-scale ionospheric irregularities basing on various geophysical parameters as a first step towards developing a model from Swarm 16-Hz in situ measurements of electron density. As it has been stated in the manuscript, The dependence of ionospheric irregularities on the geophysical parameters remains a problem in modeling their variation for predictive purposes (Yizengaw and Groves, 2018). Therefore, further global-scale studies on the distribution characteristics of ionospheric irregularities and their dependence on various factors are still necessary. Currently, the team involved in this paper are carrying out further research involving modeling of amplitude scintillation from Swarm 16 Hz measurements using the WAM model.

– Here, we mostly dealt with small scale ionospheric irregularities which are relevant for L-band scintillations. Some previous studies (e.g, Xiong et al., 2016; Wan et al., 2018) developed an algorithm to determine depletion amplitudes and they showed that large depletion amplitudes of EPBs are relevant for causing radio signal disruptions. On the other hand, other previous studies have also stated the relevance of the small-scale $N_e$ structures for causing L-band scintillations (e.g, DOI 10.3389/fphy.2014.00015,DOI 10.1002/2016JA023222, DOI 10.1007/s10509-018-3303-4). A point of reconciliation may be retrieved from Sharma et al. 2018, DOI 10.1007/s10509-018-3303-4 who suggested that the small scale structures might be abundantly available inside the largely depleted EPBs which becomes the cause of L-band scintillation. From the method used in our study, we mainly focus on irregularities of wavelength of about 15 km along the satellite track and this wavelength is within the range of applicable Fresnel scales which is theoretically relevant as a cause of L-band scintillations and therefore, here we characterize small scale ionospheric irregularities.

– We also directly compared the relative and absolute small scale irregularity perturbations of $N_e$ from a meridional point of view using data from Swarm 16 Hz over the low latitude region. As far as we know, Huang et al 2014 used the 512 Hz C/NOFS satellite's measurements of ion density and found that when the relative and absolute density disturbances are used independently, the likelihood of irregularities occurring and their variation with local time differ. However, the C/NOFS satellite was in a low tilt orbit, so the bubbles were sampled zonally. Important differences basing on latitudinal distribution of ionospheric irregularities using different criteria could not be addressed by Huang et al 2014. The polar-orbiting Swarm satellites sample bubbles in a meridional direction and they give an opportunity to check the difference in the latitudinal distribution of irregularities using different identification criteria. Ionospheric irregularities exhibit anisotropy with respect to the magnetic field, which is very pronounced at short wavelengths (tens of meters, doi:10.1029/95JA03098), and expected to be signficant also at km scales. C/NOFS and Swarm cover both different latitudinal distributions and aspect angles with respect to the

magnetic field. Also, Chartier et al 2018 used $N_e$ and Total Electron Content (TEC) measurements made by the LPs and GPS, respectively on board Swarm to test the relative and absolute perturbations in the detection of polar cap patches. In terms of seasonal distribution, they observed discrepancies between the two methods with relative disturbances showing more patches in winter than in summer. However, the study presented by Chartier et al 2018 were confined to high latitudes. Therefore, in this study we extend the direct comparison of relative and absolute irregularity perturbations of $N_e$ to the low latitudes using Swarm in the merdional direction.

– As an aspect which is possibly relevant for especially the small-scale irregularities we have also analyzed the along-track gradient of Ne (as a proxy of the pressure gradient).

– We have highlighted the above points in the revised manuscript. (please see the detailed descriptions at Pg. 2 L. 8-11, Pg.3 L. 28-33,Pg.4 L. 1-5, Pg. 9 L.7-16, )

**Comment:**I do not think, that the 16 Hz shows multiple peaks, but might reveal an extended frequency content if a frequency analyses would be made.

**Response:**Basing on the reviewer's suggestion, we performed frequency analysis to make it easy to identify and evaluate structural variations in the 2 Hz and 16 Hz electron density data. The type of frequency analysis we carried out is the Fourier transformation and the results is shown in the right panel of the figure below. The right panel shows the amplitude of the measured oscillations vs. the frequency in logarithmic scale. It can be seen that the 16 Hz data has an extended

frequency range compared to the 2 Hz data. One can see that the linear spectral index continues almost one order of magnitude to higher frequencies which corresponds to smaller scale structures. Therefore, the frequency spectrum is significantly different between 16 and 2 Hz data. The extended frequency should be because of the high resolution of the 16 Hz data and in the left panel multiple small scale density depletions can be observed in the 16 Hz data. To make it clear, the sentence in the revised manuscript has been rephrased to,
However, smaller scale electron density depletions in $N_e$ cannot be verified with the low resolution 2 Hz data as shown in the zoomed-in sections in Fig.1. (see Pg. 5 L.8-9)

**Comment:**Answer to comment 6 of reviewer 1 is not very relevant. It is not important where the irregularity occurs, nor it is the sampling rate difference between Chartier et al. and the submitted paper, but Chartier et al. showed that the relative variation is elevated in regions of lower background electron density. This is re-shown by the submitted paper.

**Response:**We acknowledge that Chartier et al 2018 showed that the relative variation is elevated in regions of lower background electron density for polar cap patches. As far as we know, Chartier et al 2018 used $N_e$ and Total Electron Content (TEC) measurements made by the LPs and GPS recorded at frequency of 2 Hz, respectively on board Swarm to test the relative and absolute perturbations in the detection of polar cap patches. Hence, the study presented by Chartier et al 2018 was confined to high latitudes to examine the polar cap patches. In the present manuscript, we extended the direct comparison of relative and absolute irregularity perturbations of $N_e$ to the low latitudes where the distribution of irregularities is quite different from that at the polar region. In addition, Chartier et al 2018, presented patch occurrence rates during the period of study. However, Wan et al 2018 showed that the highest occurrence of rate does not always coincide with the largest depletion amplitude. Therefore, in this study we presented magnitudes of the ionospheric irregularities which is different from what was presented by Chartier et al 2018. Also, the algorithms adopted by Chartier et al 2018 utilized median smoothing filters of large windows to remove small-scale structures. As mentioned in the manuscript, High-resolution data enables smaller scale structures to be identified in electron density (Nishioka et al., 2011). Therefore,in the current manuscript, we quantified small scale structures using the high-resolution data which are relevant for L-band scintillation and this is different from the algorithm used by Chartier et al 2018 which removed small-scale structures.

In the current manuscript, we show that the relative variation is elevated in regions of lower background electron density, but at low latitudes and using 16 Hz faceplate electron density data in the meridional direction. The sampling rate difference and the latitudes where irregularities occur are important factors which have been known to determine the scale size of irregularities observed and to affect the distribution of occurrence of ionospheric irregularities, respectively. Therefore, it was relevant to extend the direct comparison relative and absolute perturbation to the low latitude region in the meridional direction using the 16 Hz faceplate electron density data.

– We have highlighted the above points in the revised manuscript. (please see the detailed descriptions at Pg. 4 L. 1-4, Pg.2 L. 34,Pg.11 L. 10-12, Pg. 9 L.7-16, )

**Comment:**Linear correlations between bubble detections and Kp in response to reviewer 2 are irrelevant. That the two quantities are not linearly correlated has been shown by Palmroth et al and Stolle et al. . The presented results show the same trends as in the mentioned papers. Among other things.

**Response:** We acknowledge that Palmroth et al (2000) and Stolle et al (2006) examined the correlation between occurrence of ionospheric irregularities and Kp. Palmroth et al. (2000) found a negative correlation between Kp and pre-midnight plasma depletions and a positive post-midnight correlation. Stolle et al. (2006) checked the response of the occurrence of ionospheric irregularities to geomagnetic activity using magnetic field measurements made by CHAMP and they observed a weak relation between the occurrence of irregularities and the Kp index. However, both studies used data sets that are in crucial aspects different than the one used in the present manuscript. The DE-2 data used by Palmroth et al. (2000) were on average at higher altitude (about 600 km). The lower orbit of Swarm can explain why we find a better correlation with Kp than Palmroth et al. (2000). The study by Stolle et al. (2006) used the magnetic field instead of directly the electron density, relying on the diamagnetic effect. The methods used by Palmroth et al (2000) and Stolle et al (2006) detect only strong irregularities, while the present manuscript includes also weak irregularities which cannot be detected reliably in the magnetic data alone. Therefore we don't agree, that the correlations with Kp using the Swarm data are "irrelevant".

In the presented manuscript we aimed at showing that the electron density measurements of Swarm faceplate can be applied to examine the characteristics of ionospheric equatorial irregularities at sub - kilometers scale lengths. We also obtained a weak correlation between Kp and std($dNe$) (std($dN_e$)/$N_e$) for small scale ionospheric irregularities which confirms the capability of the Swarm 16 Hz $N_e$ data. We discuss the Kp correlation outputs in this manuscript as additional evidence that the Swarm 16 Hz $N_e$ data is able to capture small scale ionospheric irregularities and also reproduce the irregularities dependence on Kp as in previous studies. We prefer to maintain the obtained results to show that Swarm 16 Hz data is a reliable data set because it produced similar observations as in earlier studies especially with regards to correlation between irregularities and Kp.

**Response to Reviewer 2 Comments**

Second review report for "Traits of sub-kilometer F-region irregularities as seen with the Swarm satellites" authored by S. Aol et al.

**Comment:** Authors had answered all the comments and modified the paper properly. Still, I have a minor suggestion to add the reply of my second comment in the manuscript to make it more robust. And I suggest the authors to do a clearer correlation analysis between std(dNe) or std(dNe)/Ne with Kp, more criteria like local time, seasons, solar flux could be set. If there are still no correlations, authors could simply mention the results in the manuscript.

**Response:**

- We thank the reviewer for acknowledging our efforts in modifying the manuscript in an appropriate manner and we also appreciate the reviewer for the valuable comments and suggestions which have improved the quality of this manuscript.

- Concerning the reply to the reviewer's second comment, the suggestion to include it in the manuscript has been taken and the description has been incorporated in the revised manuscript. (see Pg. 9 L. 6-16)

- As suggested by the reviewer, a correlation analysis was done between std($dN_e$) (std($dN_e$)/$N_e$) with Kp and the results are presented below for only Swarm A: The Swarm passes were categorized into different seasons i.e.,

[Figure]

March Equinox, June Solstice, September Equinox and December Solstice. For each season, the Swarm passes were further categorized into pre-midnight (18:00 LT - 24:00 LT) and post-midnight (24:00 LT - 06:00 LT) local times, respectively. Generally, the results show a weak negative correlation between std($dN_e$) and Kp close to zero, irrespective of the method used to quantify the level of equatorial ionospheric irregularities and the latitude range. Close inspection shows that the correlation between std($dN_e$) and Kp is negative at the EIA belts in all seasons and

for all local time categories, except in June Solstice where the correlation is a weak positive pre-midnight. The pre-midnight and post-midnight dependence of the $std(dN_e)$ on Kp contradicts with conclusions drawn by Palmroth et al. 2000, DOI 10.1029/1999JA005090. Palmroth et al. 2000 found a negative correlation between Kp and the plasma depletions for pre-midnight events and a positive correlation for post-midnight events up to a Kp value of 4. The correlations obtained here are marginal in both time sectors and we cannot regard them as significant. The correlation presented for post-midnight may be affected due to the low data availability.

– In general, the correlation between Kp and $std(dN_e)$ $(std(dN_e)/N_e)$ is still a weak negative and we have therefore highlighted this general conclusions in the manuscript. (see Pg. 17 L. 9-11)

**Minor comments**

**Comment:**(1) P8, Line 10: should be "..occurrence at post-midnight..".

**Response:**The sentence has been modified as reviewer suggests. (see Pg.8 L.11)

[revised manuscript text omitted]

---

## Author Response (AR3)

**"Traits of sub-kilometer F-region irregularities as seen with the Swarm Satellites"**

by Aol et al.

Dear Dr. Petr Pisoft,

Thank you once again for your letter. The minor revisions suggested by the reviewers for our manuscript angeo-2019-50 are highly appreciated. In enhancing the quality of the paper, all the minor revisions we received have been taken into consideration and we present our response to each of them individually below. A marked-up manuscript version has also been embedded at the end of this document. We trust that our manuscript is acceptable for Annalese Geophysicae with the constructive referee comments and suggestions implemented.

Best regards

Sharon Aol

For the convenience of the referees we have repeated in the response the relevant comments and then given text highlights in the revised manuscript in blue.

**Response to Reviewer 4 comments**

**Comment:** The paper provides an in-depth analysis of small scale ionospheric irregularities as measured in-situ by the Swarm mission, making use of the high cadence data from the faceplate of the Electric Field Instrument. The climatological characteristics of the new data that are presented are similar to those of earlier data sources which had lower cadence and/or different geometrical sampling characteristics. Nevertheless, the extensive analysis of the EFI face plate data extends knowledge to smaller spatial scales along the Swarm orbits and therefore warrants publication.

**Response:** We are thankful for this positive comment and for warranting our paper for publication.

**Minor Comments**

I have only very few minor technical comments.

**Comment:** For instance, readers not familiar with Swarm will not know about the "faceplate", and this term is not well represented in high-level documentation on the mission. Although it is explained in Section 2, I would advise to mention the relation between the faceplate, TII, and EFI already in the abstract and introduction.

**Response:** Basing on the reviewer's advise, the sentence in the abstract (Pg1 Line 3-4) has been modified to:

- In this study, the distribution characteristics of ionospheric F-region irregularities in the low latitudes were investigated using 16 Hz electron density observations made by a faceplate which is a component of the Electric Field Instrument (EFI) on board Swarm satellites of the European Space Agency (ESA). (see Pg1 Line 3-4)

In addition, the following sentences have been shifted from Section 2. Data and Methods to Pg. 2 L.27-29

- Each satellite is equipped with an Electric Field Instrument (EFI) in addition to other payloads. The EFI consists of LPs, and Thermal Ion Imagers (TII) (Knudsen et al., 2017).

The following sentences have also been incuded in the introduction in Pg. 2 L. 34-35 and Pg. 3 L. 1-6 as additional description of the Swarm faceplate:

- The Swarm satellites have the capability of measuring electron density at an even higher frequency of 16 Hz by determining the current through a faceplate. This plate is electrically isolated from the satellite, negatively biased and located on the RAM side such that positive ions impact onto the relatively large surface of about $26 \times 26 \ \mathrm{cm}^2$ with super-thermal velocity due to orbital motion (Buchert, 2016). As a result, the electron density can be readily estimated from the current at a relatively high rate of 16 Hz. The faceplate acts like a planar LP, however, without the possibility of sweeps and temperature measurements. Operation of the TII requires a bias value which turned out to be unsuitable for density estimation. Therefore, the 16 Hz density estimates are only provided when the TII is inactive (Buchert, 2016).

**Comment:** Furthermore, the captions of Figures 11 and 13 refer to "black lines", which should probably be changed to "coloured solid lines".

**Response:** The captions of Figures 11 and 13 have been rewritten as suggested by the reviewer. (see captions of Figure 11 and 13 in Pg 16 and Pg 18, respectively)

**Response to Reviewer 5 Comments**

Traits of sub-kilometer F-region irregularities as seen with the Swarm satellites - Manuscript review
December 17, 2019

**General Comment:** Provided manuscript focuses on the distribution characteristics of ionospheric F-region irregularities in the low latitudes. The Authors have replied to previous comments of Reviewers and improved quality of the paper. The study uses 16 Hz electron density observations made by the faceplate on board Swarm satellites of the European Space Agency (ESA). Extensive analysis focuses on absolute (std(dNe)) and relative (std(dNe)/Ne) density perturbations, but in my opinion presented work does not fully exploits capabilities of the Swarm mission. Keeping in mind that Swarm is a mission dedicated to the Earth's magnetic field, it would be much more interesting from the scientific point of view, if the Authors could provide joint analysis of electron density and magnetic field perturbations.

It is unnecessary to perform joint analysis from the scratch, since that would highly affect the whole concept and structure of the manuscript. But Swarm provides the Level 2 data product dedicated to plasma irregularities, the ionospheric bubble index. Please visit Swarm repository to download the data:
`https://swarm-diss.eo.esa.int/#swarm%2FLevel2daily%2FLatest_baselines%2FIBI` Taking example of discussion of irregularity structures observed by Swarm, it would be interesting to how these structures are classified by the plasma bubbles detection algorithm. For instance analysis of Bubble Flag in the product, could provide information on accompanied fluctuations in the Swarm magnetic field registrations.

**Response:**

- We thank the reviewer for acknowledging our efforts in modifying the manuscript in an appropriate manner.
- The reviewers suggestion to exploit further the Swarm satellite data-sets by providing joint analysis of electron density and magnetic field perturbations is an important/interesting line of study which needs to be examined further. The team involved in the current manuscript is also developing an interesting concept worth another manuscript and it involves examining the Swarm electron density and magnetic field data variations. Sample result is presented here: It is observed that fluctuations in magnetic field correspond to regions of electron density perturbations. As

rightly mentioned by the reviewer, performing the suggested joint analysis would affect the whole concept of the

current manuscript, since this study aimed at checking the capability of the Swarm 16-Hz faceplate electron density measurements for small-scale ionospheric irregularity observations. We are definitely interested in the analysis of magnetic perturbations in relation to observed electron density variations and we have taken the reviewer's suggestions as a point for consideration for future study. Thank you for the suggestion.

– Concerning the reviewer's suggestion of analyzing the IBI, it is an important and interesting suggestion. Basing on the reviewer's suggestion, we followed the link provided and checked on one event for Swarm A and 2014-10-06 in comparison with electron density variations and the results are presented here in Figure 0. The IBI clearly captures the fluctuations in electron density. IBI provides information on EPD climatology itself as well as on the disturbance level of the magnetic field data by taking both electron density and magnetic field measurements into account.

[Figure]

**Figure 0.** Swarm A electron density variations on 2014-10-06 in comparison with IBI

Examining further the IBI, electron density and the magnetic field data is an important/interesting line of study. However, for the current manuscript, we aimed at checking the capability of the Swarm 16-Hz faceplate electron density measurements for small-scale ionospheric irregularity observations. We would also be interested in the analysis of IBI in relation to observed electron density variations and we have taken the reviewer's suggestions as a point for consideration for future study. Thank you for the suggestion.

[revised manuscript text omitted]